# Review of methodological considerations and recommendations for mapping remote glaciers from aerial photography surveys in suboptimal conditions

Dorota Medrzycka[1], Luke Copland[1], Laura Thomson[2], William Kochtitzky[3], and Braden Smeda[1]

[1]Department of Geography, Environment, and Geomatics, University of Ottawa, Ottawa Ontario, Canada
[2]Department of Geography and Planning, Queen's University, Kingston Ontario, Canada
[3]School of Marine and Environmental Programs, University of New England, Biddeford Maine, USA

**Correspondence:** Dorota Medrzycka (dorota.medrzycka@uottawa.ca)

**Abstract.** Structure from motion (SfM) photogrammetry coupled with multiview stereo (MVS) techniques are widely used for generating topographic data for monitoring change in surface elevation. However, study sites on remote glaciers and ice caps often offer suboptimal conditions, including large survey areas, complex topography, changing weather and light conditions, poor contrast over ice and snow, and reduced satellite positioning performance. Here, we provide a review of methodological considerations for conducting aerial photography surveys under challenging field conditions. We generate topographic reconstructions, outlining the entire workflow, from data acquisition to SfM-MVS processing, using case studies focused around two small glaciers in Arctic Canada. We provide recommendations for the selection of photographic and positioning hardware and guidelines for flexible survey design using direct measurements of camera positions, thereby removing the need for ground control points. The focus is on maximising hardware performance despite inherent limitations, with the aim of optimising the quality and quantity of the source data, including image information and control measurements, despite suboptimal conditions.

## 1 Background

In the last couple of decades, digital photogrammetry techniques have become more accessible, enabling the rapid acquisition of high resolution topographic information at low cost (e.g., Westoby et al., 2012; Fonstad et al., 2013; Smith et al., 2016a; Anderson et al., 2019). The use of structure from motion (SfM) algorithms coupled with multiview stereo (MVS) computer vision methods has become widespread in glacier monitoring as a tool for mapping ice extent and surface topography, and for quantifying ice volume change through time (e.g., Mölg and Bolch, 2017). As an image processing technique, SfM relies on matching features detected on multiple overlapping 2D images to estimate camera motion and the position of these given features in a 3D point cloud. MVS algorithms complement the workflow, refining the initial point cloud to generate high resolution topographic reconstructions. In glacier studies, air photos acquired from single and repeat airborne surveys can be used to create digital elevation models (DEMs) and orthophotos representing the glacier surface. These reconstructions are useful tools for tracking ice motion (Immerzeel et al., 2014; Chudley et al., 2019; Jouvet et al., 2019), monitoring calving dynamics (Ryan et al., 2015; Jouvet et al., 2017, 2019), quantifying elevation changes and surface ablation (Bash et al., 2018;

Medrzycka et al., 2023), and mapping surface topography (Smith et al., 2016b), crevasse patterns (Thomson and Copland, 2016), supraglacial drainage networks (Rippin et al., 2015; Bash and Moorman, 2020), and debris cover (Kraaijenbrink et al., 2016, 2018).

The flexibility of the SfM-MVS technique implies that methods, including survey design, processing steps, and settings used, differ between studies. As there is no consistent methodology, efforts have been made towards establishing general guidelines for a standardised workflow with rigorous error reporting strategies for the use of SfM in the geosciences (e.g., James et al., 2019). A number of publications summarise best practices in terms of survey design and data acquisition for optimal results under typical conditions, including the choice of airborne platform, photographic and positioning hardware, image network geometry, and image capture settings (e.g., Eltner et al., 2016; Smith et al., 2016a; Mosbrucker et al., 2017). In theory, proper planning and careful survey design can minimise measurement errors, but in reality, several interacting factors in the field can influence the quality of final outputs, making it difficult to predetermine expected errors and identify the factors with the largest impact on accuracy. This is especially crucial where field data acquisition is performed in suboptimal conditions, and where careful survey planning is impractical or impossible as is often the case when surveying glaciers in remote locations.

Conditions are often suboptimal for conducting air photo surveys on glaciers due to large survey areas, rugged terrain, changing light conditions, lack of contrast over snow-covered glaciers, poor satellite reception, and proximity to the magnetic poles that limits the operation of remotely piloted aircraft systems (RPAS). Here, we present a set of guidelines for flexible survey design in suboptimal conditions, where the aim is to optimise data acquisition and ultimately minimise the impacts of adverse field conditions on model results. We illustrate our recommendations using data from two air photo surveys flown from helicopters in the summers of 2018 and 2019, and build centimetre-scale DEMs and orthomosaics for two small ice masses in the Canadian Arctic. This work is largely based on the PhD thesis of Medrzycka (2022) where data processing steps and the entire SfM-MVS workflow are described in full detail. In the following sections, we:

1. Review some of the inherent challenges to designing air photo surveys while optimising flexibility and efficiency of the data acquisition process in remote regions, and discuss key elements in hardware selection for imaging and positioning systems.

2. Present the field data acquisition process for the two case studies, including survey setup, global navigation satellite system (GNSS) control measurements, and image capture settings.

3. Outline processing steps to optimise image and positioning data prior to generating topographic reconstructions, including image enhancements for increased contrast and details, and camera position estimates and associated uncertainties from raw GNSS data.

4. Summarise the main outputs from the SfM-MVS workflow and discuss their relevance for field surveys and value for glacier change detection studies.

5. Provide recommendations for optimising raw data and the quality and reliability of final products derived from aerial photography surveys in suboptimal conditions.

## 2 Survey design & hardware selection

### 2.1 Airborne platform

Remotely piloted aircraft systems are becoming an increasingly common tool for performing aerial surveys, with a major advantage being the much lower acquisition and operation costs of remotely piloted aircraft compared to the cost of chartering crewed aircraft. However, both fixed-wing and multi-rotor remotely piloted aircraft have limitations, including limited payload and horizontal and vertical range. Large survey areas, complex topography, rugged terrain, and challenging weather patterns (including high winds, low temperatures, and atmospheric icing conditions) also represent major obstacles to RPAS operations in glaciated regions (e.g., Gaffey and Bhardwaj, 2020). Perhaps the most significant limitation is the performance of control systems and autonomous navigation: RPAS typically rely on magnetic sensors for attitude determination during navigation, but these can be severely compromised in proximity to the magnetic poles. This issue has been particularly problematic in locations close to the magnetic poles, and our own tests with several different RPAS resulted in almost immediate crashes following the failure of the magnetic orientation. Crewed aircraft, particularly helicopters, are therefore the only viable option to survey large glaciers in remote polar locations. Major advantages of crewed aircraft include their high payload (often up to $\sim$1000 kg), which allows for the use of larger and higher performance photographic and positioning equipment than is typically possible with remotely piloted aircraft, and their greater flight range.

### 2.2 Imaging system

Image quality primarily depends on the camera/lens combination and its ability to resolve fine details in all environmental conditions, regardless of ambient light or surface properties. Imaging systems with interchangeable lenses, including digital single lens reflex (DSLR) and mirrorless cameras, are more flexible in terms of hardware selection and image capture settings, and therefore offer more control over data acquisition than compact cameras fitted on most consumer-grade remotely piloted aircraft. In addition, the reduced sensor size of smaller compact cameras often translates into smaller pixel size, a key factor impacting the signal to noise ratio and the level of detail available in shadows and highlights (Rowlands, 2020, chap. 5, p. 2). Along with focal length, sensor size and pixel count also define the ground sampling distance (GSD) and therefore the spatial resolution and maximum achievable precision of the resulting topographic reconstruction. High resolution imaging systems such as full frame DSLR cameras allow for increased flight height while ensuring sufficiently high spatial resolution. This maximises survey range and enables more efficient coverage of larger areas with fewer flightlines and/or overlapping images (Smith et al., 2016a).

In photogrammetry, the key for a high-performance setup dedicated to precision measurements is matching the camera with a high optical quality lens to ensure maximum image resolution and sharpness, high sensitivity and dynamic range, and to minimise aberrations and distortions. This section expands on four key elements to consider when selecting an imaging system, namely the camera sensor, the lens, signal processing, and camera shutter mechanism. Where relevant to illustrate the importance of imaging hardware choice, we make comparisons between the Nikon D850 DSLR camera (35 mm, 45.7 megapixels, with NIKKOR AF-S 24mm $f$/1.8G ED) that we use for our work, and other commercially available options.

**Table 1.** Comparison of common sensor sizes and respective resolution limits. FF: full frame; MF: medium format. The Nyquist limit, defined in units of cycles per mm (cy mm$^{-1}$), is related to pixel pitch ($\rho$) by $0.5/\rho$, and is equivalent to one half of the sensor cut-off frequency. The Nikon D850 fits in the FF category of newer sensors with relatively higher pixel count than the more standard 20–24 Mpx.

| Sensor format | Sensor size (mm) | Resolution (Mpx) | Pixel pitch (μm) | Nyquist limit (cy mm$^{-1}$) | Focal multiplier |
|---|---|---|---|---|---|
| 1" | 13.2 × 8.8 | 20 | 2.4 | 208 | 2.7 |
| APS-C | 23.5 × 15.6 | 24 | 3.7 | 135 | 1.5 |
| FF | 36 × 24 | 24 | 6 | 83 | 1 |
| FF | 36 × 24 | 45.7 | 4.35 | 115 | 1 |
| MF small | 43.8 × 32.9 | 50 | 5.3 | 94 | 0.82 |
| MF large | 53.4 × 40 | 150 | 3.8 | 133 | 0.67 |

### 2.2.1 Sensor size & pixel pitch

A major advantage of full frame (FF) DSLRs is the large sensor size (36 × 24 mm) compared to a smaller crop sensor such as the common APS-C format (23.5 × 15.6 mm), or to 1"-type sensors (13.2 × 8.8 mm) on smaller cameras that fit on most remotely piloted aircraft. Sensor size generally increases pixel pitch, a metric related to the distance from the centre of one pixel on the sensor (photodiode) to the next. Pixel pitch is directly related to full-well capacity (i.e., the maximum number of electrons a photodiode can hold at saturation level), which determines the maximum signal to noise ratio. Reduced pixel size tends to increase overall sensor noise, resulting in reduced sensitivity and low-light performance. Smaller pixels require longer exposure times (Yoshida, 2006; Rowlands, 2020, chap. 5, p. 3) and can therefore be a limiting factor for surveys undertaken from a moving platform and/or in suboptimal conditions, resulting in underexposed images or motion blur. This implies that a larger number of megapixels is not necessarily a plus since, to maintain an equivalent resolution, smaller sensors must have smaller pixels (Table 1). However, as sensor design is a constantly evolving technology, two sensors of different generations with similar size and pixel count might not reach the same level of performance, with newer generation designs often outperforming older ones.

Recent developments in CMOS (complementary metal oxide semiconductor) technology, and the introduction of backside illuminated (BSI) sensors in the 2010s, for example, have resulted in significant improvements in low-light sensitivity particularly for small pixel sensors. Unlike standard front illuminated sensors, in BSI sensors, the wiring has been moved behind the photodiode, increasing the photosensitive detection area (the photosite) and improving full-well capacity (Ohta, 2020). Full-well capacity has an additional effect on dynamic range (i.e., range of luminance values in an image) which determines performance in high contrast environments (Yoshida, 2006; Rowlands, 2020, chap. 2, p. 6). This is especially important where the intensity difference between a bright glacier surface and the darker bare ground along the margins is amplified by low sun angles and long shadows on the surface.

Most consumer cameras record 8–12 bits per channel, while higher performance imaging systems use higher resolution 12–14 bit depth data. The raw digital information can be saved without compression (preserving the original bits) in RAW image format, and/or converted through nonlinear encoding and compressed into standard 8-bit image files, typically TIFFs or JPEGs. Raw bit depth dictates the number (and size) of quantisation steps available for encoding digital data, which directly translates to tonal precision, or the number of tonal levels in an image (Yoshida, 2006; Rowlands, 2020, chap. 2, p. 7). High bit depth therefore allows subtle tonal variations to be more faithfully resolved, which becomes particularly crucial in low contrast conditions and uniform surfaces such as in snow covered areas.

The modulation transfer function (MTF) is a performance metric used to describe the resolving power of an imaging system, or the ability of a camera and lens combination to reproduce fine detail at characteristic spatial frequencies. Pixel pitch (and the detection area) is the primary factor dictating the theoretical maximum resolution (or resolving power) of an imaging system. Resolution, which stands for spatial frequency, is typically defined in units of cycles (or line pairs) per mm (cy mm$^{-1}$ or LP mm$^{-1}$), corresponding to the number of light/dark line pairs that can be properly resolved per unit on the focal plane (the sensor). The maximum achievable resolution is limited by the detector cut-off frequency, equivalent to twice the Nyquist frequency (0.5 cy px$^{-1}$) at which point the sensor MTF drops to zero and no further information can be resolved (Palum, 2009; Rowlands, 2020, chap. 3, p. 38). Above the Nyquist frequency, an imaging system is unable to properly resolve fine details without introducing aliasing artefacts (also called moiré). With the exception of medium and large format professional cameras, many high resolution cameras were, until recently, equipped with an optical lowpass filter (or antialiasing filter) in order to minimise aliasing noise when capturing fine repetitive detail (e.g., lines or dots) above the detector Nyquist frequency. The filter cuts high spatial frequency information essentially blurring the image and causing some resolution loss (Palum, 2009). Since patterns causing moiré are virtually absent in nature, lowpass filters are only a downside for aerial photogrammetry applications where maximum image sharpness is key. Higher pixel count and smaller pixel pitch reduce the chance of moiré, therefore lowpass filters are generally absent on cameras with smaller (non-FF) sensors and are increasingly being abandoned on newer DSLRs and mirrorless cameras as sensor resolution increases (Palum, 2009).

To illustrate the importance of sensor MTF when selecting a camera, consider the Nikon D850 with a pixel pitch ($\rho$) of 4.35 μm, which places the detector cut-off frequency ($\nu_c = 1/\rho$) at 230 cy mm$^{-1}$, and the equivalent Nyquist limit at 115 cy mm$^{-1}$ (Fig. 1). This is roughly 30 % higher than the 166 cy mm$^{-1}$ cut-off frequency of a standard lower resolution FF camera such as the Nikon D750 with 24 megapixels and a pixel pitch of 6 μm. As sensor MTF is the product of detector and filter MTFs, the addition of the lowpass filter on the D750 further limits maximum system resolving power and lowers the cut-off to the Nyquist frequency of 83 cy mm$^{-1}$ (Fig. 1). This implies that system resolving power increases with decreasing pixel size, although system cut-off frequency is ultimately determined by whichever component (detector/sensor or lens) has the lowest cut-off (Palum, 2009; Rowlands, 2020, chap. 3, p. 49).

### 2.2.2 Lens performance

Lens MTF response is evaluated in terms of relative contrast as a function of frequency and depends on two main factors: diffraction and aberrations. In theory, a perfect lens is said to be diffraction limited (i.e., resolving power depends on diffraction

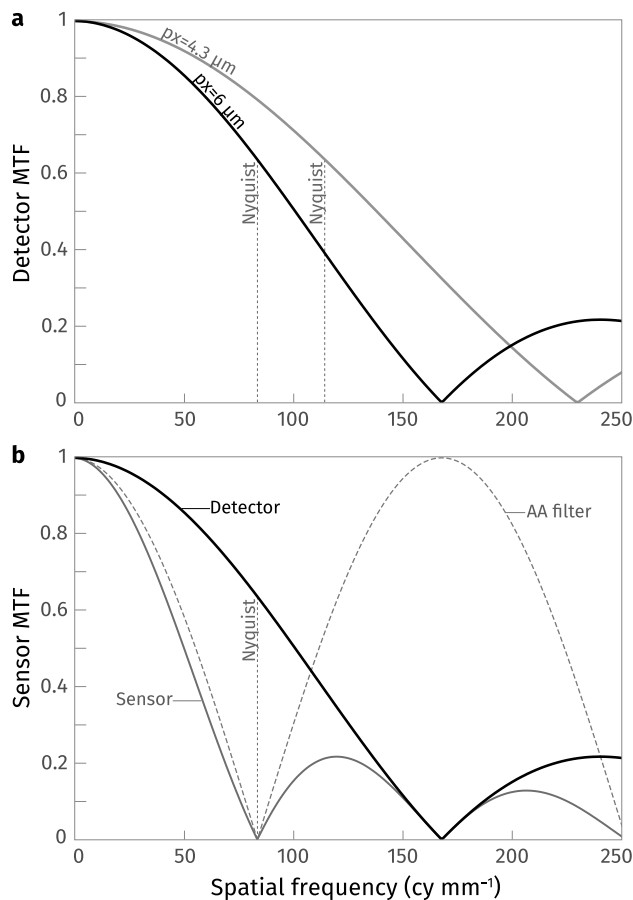

**Figure 1.** (a) Detector frequency response as a function of pixel pitch alone for a full frame high resolution (45.7 megapixel) sensor (Nikon D850; grey line), and a full frame lower resolution (24 megapixel) sensor (Nikon D750; black line). Here, the pixel detection area is assumed to be equivalent to pixel pitch $\rho = 4.35$ μm (D850) and $\rho = 6$ μm (D750). This places the detector cut-off frequency ($\nu_c = 1/\rho$) at 230 cy mm$^{-1}$ (D850) and 166 cy mm$^{-1}$ (D750), and the Nyquist frequency at 115 cy mm$^{-1}$ (D850) and 83 cy mm$^{-1}$ (D750). (b) Sensor frequency response for the D750 with the addition of a lowpass (antialiasing: AA) filter. The sensor modulation transfer function (MTF) is the product of both detector and antialiasing filter MTFs, with the cut-off coinciding with the Nyquist frequency (83 cy mm$^{-1}$). Notice how the addition of the AA filter also lowers the sensor MTF below the Nyquist frequency. Detector and AA filter MTFs are modelled with: $MTF_{det} = sin(\pi\nu\rho)/(\pi\nu\rho)$ and $MTF_{AA} = cos(\pi\nu\rho)$ (Rowlands, 2020, chap. 5, p. 37).

alone), however, in reality, all lenses suffer from various imperfections which combine to lower the real-world lens MTF (Ray, 2002, pp. 145–8).

Diffraction occurs as light passes through a circular aperture, causing waves to bend around the edge and preventing them from converging to a single point on the focal plane. This produces an Airy disk pattern (consisting of a central spot surrounded by concentric circles), which causes blurring and sets a fundamental limit on the maximum achievable resolution determined

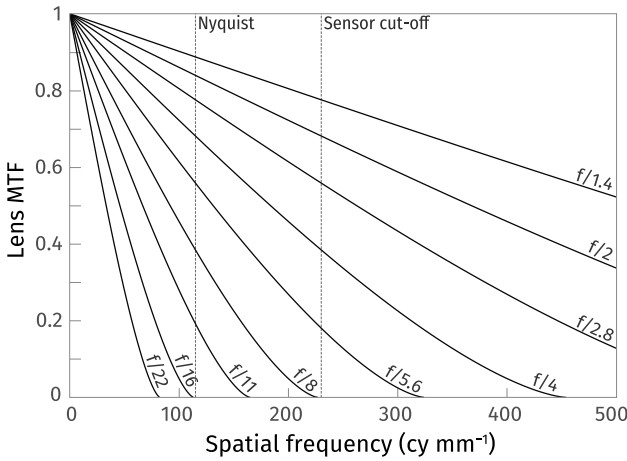

**Figure 2.** Lens modulation transfer function (MTF) as a function of spatial frequency for a diffraction limited (aberration-free) lens with a circular aperture at various f-numbers. The cut-off frequency (or in this case the diffraction limit: $\nu_c = 1/(\lambda N)$) is calculated for green light with a wavelength of $\sim$550 nm. The sensor cut-off and Nyquist frequencies (two dotted vertical lines) are those calculated for the Nikon D850, with a pixel pitch of 4.35 μm and no antialiasing filter. Lens MTF is modelled with: $MTF_{diff} = \frac{2}{\pi}(cos^{-1}(\nu/\nu_c) - (\nu/\nu_c)\sqrt{1-(\nu/\nu_c)^2})$ (Rowlands, 2020, chap. 3, p. 31).

by sensor design. Diffraction is a function of the wavelength of the incoming light ($\lambda$), focal length ($f$), and aperture diameter ($D$), with $f/D$ giving the f-number ($N$), sometimes called f-stop. By definition, the diffraction limit decreases with smaller apertures (large f-numbers) (Rowlands, 2020, chap. 3, p. 28). Figure 2 shows the cut-off frequency ($\nu_c = 1/(\lambda N)$) and lens MTF as a function of spatial resolution ($\nu$) for an aberration-free lens with a circular aperture. For example, taking the middle wavelength of natural (green) light at $\sim$550 nm, and an aperture of $f/4$, puts the diffraction limited frequency at 455 cy mm$^{-1}$. As this is higher than the detector cut-off frequency of 230 cy mm$^{-1}$ calculated for the D850, the system resolving power is limited by sensor resolution. However, stopping down to $f/8$ reduces the limit to 227 cy mm$^{-1}$ at which point the system resolving power becomes diffraction limited (Fig. 2). Taking into account various aberrations further lowers lens MTF and therefore system resolving power.

Aberrations are inherent to lens design and occur where light fails to converge at a single point on the focal plane or along the optical axis. Chromatic aberrations result from different wavelengths focusing at different positions on the focal plane causing colour fringing (Ray, 2002, pp. 112–4), most often visible as purple artefacts along high-contrast boundaries. Monochromatic aberrations include spherical aberrations, coma, astigmatism, and field curvature, all of which cause blur and contrast reduction. With the exception of spherical aberrations which affect the entire image, the other types vary with field position and intensify as a function of distance from the optical axis (Ray, 2002, pp. 82–3). Lens MTF response therefore follows the same pattern and contrast reproducibility is typically highest at the centre of the frame and degrades towards the edges. While lenses tend to lose significant contrast to diffraction effects at small apertures (large f-numbers), aberrations are amplified at large apertures

(small f-numbers). Therefore, the optimum aperture for maximum performance hovers around two to three stops up/down from their maximum/minimum aperture, respectively, typically around $f/4$ to $f/5.6$ (Ray, 2002, p. 147).

Ultimately, system MTF is the product of the transfer functions of all individual components combined. The system MTF has implications for determining the spatial resolution of any data products derived from images captured at a given aperture. Spatial resolution is often thought of as equivalent to the ground sampling distance (GSD), or distance between two consecutive pixel centres on the ground, calculated with:

$$GSD = \frac{H \times \rho}{f} \tag{1}$$

where $H$ is height above ground level (a.g.l.), $\rho$ is pixel pitch, and $f$ is focal length. The GSD assumes that the resolution limit coincides with the spatial frequency where the sensor MTF drops to zero, omitting the influence of imaging optics on system resolving power. Besides, the zero MTF limit is rather theoretical since, in reality, the effective cut-off frequency where resolution is sufficiently high for an image to retain any useful information is around an MTF of 9 %, as implied by the Rayleigh criterion (Rowlands, 2020, chap. 5, pp. 32–3).

In aerial (and satellite) photography, ground resolved distance (GRD) refers to the smallest resolvable detail on an image, given the limitations of the imaging system, including diffraction effects. From a spatial domain point of view, diffraction becomes gradually more visible as the diameter of the Airy pattern (also called circle of confusion) on the focal plane increases relative to pixel pitch, and is generally tolerable up to 1.5 pixels. Based on the Rayleigh criterion, the effective cut-off frequency corresponds to the point where two Airy disks are no longer resolvable, that is, where the distance between the centres of two disks is equal to their radius: $1.22\lambda N$ (Ray, 2002, pp. 145–6). In other words, the system becomes diffraction limited when the diameter of the circle of confusion reaches a size of twice the pixel pitch. Therefore, taking into account diffraction effects should place a limit to the GRD equivalent to the diameter of the circle of confusion, given by:

$$GRD = 2.44 \times H \times \frac{\lambda N}{f} \tag{2}$$

which varies with different f-numbers (Fig. 3). With the Nikon D850 and a 24 mm lens, $\rho = 4.35$ μm, $f = 24$ mm, and taking $H = 500$ m, the GSD is 90 mm. With an aperture $N = f/4$ and $\lambda = 550$ nm, the GRD is 112 mm ($\sim 1.2 \times$ GSD), meaning, despite some amount of image degradation, the optics are not limiting the GSD. However, take $N = f/8$ and the GRD becomes 224 mm ($\sim 2.5 \times$ GSD) and so the base GSD of 90 mm cannot be properly resolved. Naturally, this still represents an idealised scenario, applicable to a system with perfect optics. In general, smaller consumer cameras are more likely to have lower quality optics with greater wavefront aberrations further lowering the resolving power, especially so at larger apertures. Independently from the quality of the system itself, defocus or motion blur due to camera motion combined with slow shutter speeds, can put an additional limit on the GRD.

Focal length, the distance between the optical centre of the lens and the focal plane, defines the angular field of view and object magnification for a given lens. Wide angle lenses (35 mm or less) are well suited for aerial photography as they allow the capture of more of the scene in a single image, compared to longer lenses with a narrower field of view (Smith et al., 2016a). Effective focal length refers to the field of view of a system taking into account different sensor sizes, calculated by applying

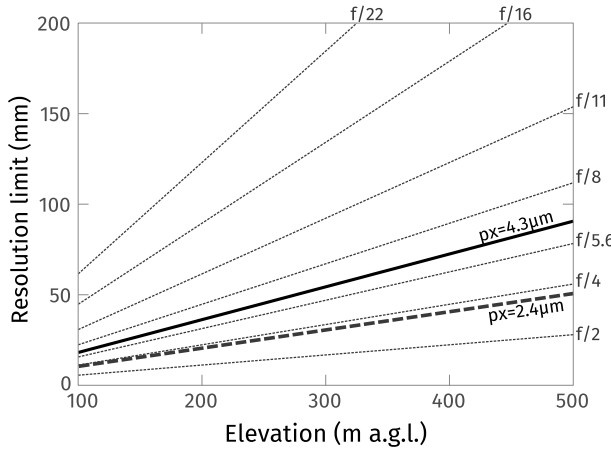

**Figure 3.** Ground resolved distance (GRD) at various f-numbers as a function of flying height above ground surface, calculated from Eq. 2. The thick lines represent the standard definition of GSD from Eq. 1 based on the sensor cut-off frequency for the Nikon D850 with a pixel pitch of 4.35 μm (solid line), and for a standard 1" sensor with a pixel pitch of 2.41 μm (dashed line). At $H = 500$ m above ground level, the D850 is (at best) able to resolve two objects 90 mm apart. Taking into account diffraction at $f/8$, the system can only resolve two objects up to 224 mm apart and so the theoretical GSD becomes limited. In comparison, the smaller 1" sensor is already limited at $f/4$.

a focal length multiplier. Sensors smaller than the standard 35 mm FF format introduce a crop factor, meaning that a 24 mm lens coupled with an APS-C sensor will have an effective focal length of 24 mm multiplied by 1.5 (or 1.6 for Canon cameras), equivalent to 36 mm on a FF (35 mm no crop) sensor. Smaller sensors therefore require wider lenses for a given field of view (Rowlands, 2020, chap. 1, pp. 36–7), the downside being that short focal lengths are more prone to distortions which can affect measurement accuracy.

Focal length and aperture also define the hyperfocal distance, corresponding to the focus distance giving the maximum the depth of field (DOF), defined as the zone of acceptable focus. The hyperfocal distance decreases with focal length and aperture, with wide-angle lenses and large f-numbers maximising the DOF. Focusing a lens at infinity places the near edge of the DOF at the hyperfocal distance (Ray, 2002, pp. 221–2) which, for an effective focal length of 24 mm at f/5.6 is 3.4 m, meaning everything falling any further will be acceptably sharp. Further closing the aperture to f/11 reduces the hyperfocal distance by about half (to 1.7 m) but also impacts system resolving power by increasing diffraction softening. In aerial photography, where the height above ground exceeds the hyperfocal distance and DOF is not a concern, selecting an aperture minimising diffraction and motion blur is preferable.

Unlike aberrations causing blur, optical distortions have no direct effect on lens MTF and cannot be compensated for by adjusting the aperture. Distortions arise from variations in magnification with field position and affect image geometry by causing straight lines to appear curved. Wide angle lenses exhibit negative (barrel) distortions which present as decreasing image magnification from the centre of the frame towards the edges, while positive (pincushion) distortions are characteristic of telephoto lenses (70 mm or above). Zoom lenses tend to display more complex distortions and a combination of both

types, transitioning from positive to negative with decreasing focal length (Ray, 2002, pp. 93–5). The amount of distortion corresponds to the difference between the real image and the theoretical (undistorted) one, often reported as a percentage of image height, and is generally less pronounced in long focuses lenses but intensifies with increasingly short focal lengths (Ray, 2002, pp. 94–5). Extreme wide angle and fisheye lenses such as those found on action cameras (e.g., GoPro), are therefore unsuitable for photogrammetry applications due to extreme distortions (Smith et al., 2016a). For aerial photography, effective focal lengths between 24 and 35 mm represent a decent compromise between field of view (62–84º) and optical quality. Prime lenses with fixed focal lengths typically have higher quality optics (and therefore lower aberrations) when compared to zoom lenses at a similar price point. This is mainly because zoom lenses require more complex designs to accommodate a range of focal lengths, while prime lenses are designed for maximum performance at a specific focal length. Since aberrations tend to vary with focal length, prime lenses are, by definition, more suitable for applications requiring repeatable measurement (Koyama, 2006). Lens performance is even more important with smaller sensors as diffraction effects become increasing problematic with decreasing pixel pitch.

### 2.2.3   Signal processing

Converting raw image data into a standard output image format (TIFF or JPEG) with a lower bit depth involves mapping a range of raw levels to a single tonal value. While this can be achieved without degrading perceived image quality by applying a nonlinear tone curve, it can also result in visible quantisation errors and posterisation, where continuous gradients appear as a series of discrete tonal transitions (or apparent banding) (Yoshida, 2006; Rowlands, 2020, chap. 2, pp. 7–8) . In addition to a lower bit depth, the JPEG format uses lossy compression which can introduce visible rounding errors and unwanted digital artefacts, most notably blocking and ringing. While block artefacts are particularly noticeable in relatively uniform areas with smooth colour gradients, ringing artefacts blur edges and impact sharp object boundaries (Yoshida, 2006). In-camera raw conversion essentially involves a series of irreversible adjustments designed to improve perceived image quality, including noise reduction, sharpening, aberration corrections, tone and colour mapping, and contrast enhancements. One of the main drawbacks of such a conversion process is the loss of luminance information and amplified quantisation errors, mainly in dark shadow areas (Rowlands, 2020, chap. 2, pp. 7–9). Selecting a camera with a high bit depth (12–14 bits) and the ability to shoot in RAW format maximises available image information and provides considerably more control over tonal and colour reproduction. Increased flexibility in postproduction allows for taking advantage of the full tonal range and retrieval of dark levels which would otherwise be lost during in-camera raw conversion (Fig. 4).

Where working with the RAW image data is not possible, in-camera processing of image information should be minimised (e.g., picture mode set to 'flat' or 'neutral', colour space to Adobe RGB) and any adjustments should be disabled in camera settings. Additional corrections applied to spatial image information can be especially detrimental to photogrammetry applications. For example, in-camera raw conversion also includes corrections targeting geometric (mainly radial) distortions which alter pixel geometry (Rowlands, 2020, chap. 4, pp.1–2) and can impact the camera calibration process performed by the camera model implemented in photogrammetry software (Brown, 1971; Fraser, 2013). Any additional image processing steps with

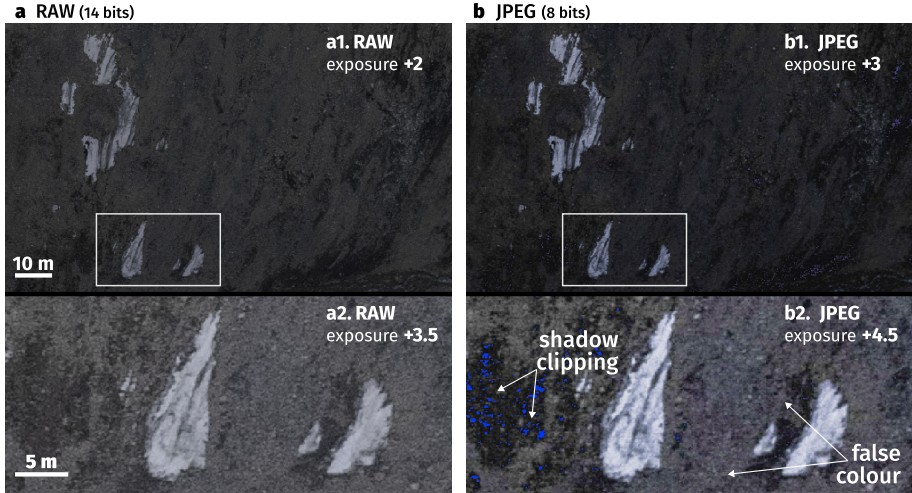

**Figure 4.** Exposure adjustments performed on two versions of a single underexposed image from the Tanquary Fiord (TF) survey (3 August 2018), captured in (a) 14-bit RAW, and (b) 8-bit JPEG formats. Both files were saved in-camera and imported into Lightroom for editing. Stronger adjustments are required for the JPEG (b1) to reach a comparable overall level of exposure and retrieve an equivalent amount of information to the RAW image (a1). With more extreme adjustments, the RAW image (a2) remains useable, while on the JPEG (b2) some information is lost in the darker shadows (bright blue pixels) and compression artefacts and false colour (purple patches) combine to degrade image quality. The RAW images were subsequently exported as 16-bit TIFFs for further processing.

the potential to modify image and/or pixel geometry should be avoided, including any optical or digital image stabilisation techniques (also called vibration reduction).

### 2.2.4 Shutter

Most DSLRs are equipped with CMOS sensors which conventionally use mechanical vertical travel focal plane shutters, also called rolling shutters. Focal plane shutters have rigid-blade curtains which travel across the sensor at a constant rate
independent of exposure time. The front curtain starts opening at the beginning of the exposure, and the rear one travels behind it after a delay equivalent to the shutter speed. The vertical travel time corresponds to the flash synchronisation speed (or X-sync speed) which is the time required for the shutter to fully open. Scanlines on the sensor array are exposed and read out sequentially over the timespan corresponding to the sum of the vertical curtain travel time and the exposure time. As a result, the exposure timing of individual pixel rows shifts across the frame. When capturing moving images this rolling shutter effect can
lead to geometric distortions. Assuming that shutter motion is orthogonal to the direction of travel, motion trace has a shearing effect where straight lines appear skewed. On the other hand, where shutter motion is parallel to aircraft motion, the resulting image will either be compressed or stretched, depending on whether the shutter curtains open and close towards or away from the direction of travel. Including the affinity and non-orthogonality coefficients in the camera calibration matrix at the image alignment stage can partially compensate for rolling shutter distortions, although it is less likely to be effective with large and

rapid changes in flight speed, direction, and height above ground. Various software, including Pix4D and Agisoft Metashape, have also implemented camera models to compensate for rolling shutter effects, estimating camera motion (translation and rotation) during exposure and modelling external orientation parameters per row of pixels on the sensor (instead of per image) (Vautherin et al., 2016). However, the performance of the correction is also sensitive to survey configuration, showing better results with more regular gridded flight patterns at relatively constant speed, and especially when combining nadir and oblique

images. When correcting for rolling shutter, simultaneously solving for the affine distortion parameters has been shown to degrade accuracy due to an overparameterisation of the model (Zhou et al., 2020).

To illustrate the impact of rolling shutter effects, consider a helicopter survey with an average flight speed of $30 \text{ m s}^{-1}$, and a vertical travel time (or X-sync speed) equivalent to 4 ms (1/250 s) for the Nikon D850, the resulting camera displacement during a single exposure is $\sim$12 cm. The effect of camera motion on the resulting image can be estimated from the pixel

displacement $\Delta_{px}$ during the curtain travel (or readout) time obtained by:

$$\Delta_{px} = \frac{u\,t\,f}{\rho H} \tag{3}$$

where $u$ is aircraft velocity, $t$ the vertical curtain travel time, $f$ the focal length, $\rho$ the pixel size, and $H$ height above ground surface. Object displacement across an image due to rolling shutter effects should ideally remain below $\sim$1 pixel. For the D850, with $t$, $f$, and $\rho$ equivalent to 4 ms, 24 mm, and 4.35 μm respectively, and the aircraft travelling at an average of $30 \text{ ms}^{-1}$

and a height of 500 m a.g.l., object displacement across the image due to the rolling shutter effect averages 1.3 pixels. It is worth mentioning that electronic rolling shutters such as those in compact consumer products typically have much slower (by an order of magnitude) effective vertical travel and readout speeds which amplifies rolling shutter effects. With all things kept equal, but using a compact camera with a typical readout speed of 30 ms, the average object displacement across an image would be closer to 10 pixels.

An alternative for dealing with this issue is to use a global shutter which exposes the full pixel array at the same instant. Electronic global shutters are mainly used with lower-resolution CCD (charge coupled device) sensors, included in compact cameras and older DSLRs, which are prone to smearing and diffraction effects (blooming), degrading image quality. Large format metric cameras developed for scientific imaging applications, and those specifically engineered for airborne operations, are often equipped with mechanical global shutters and full frame CCD sensors which offer superior performance, including

high dynamic range, but are highly specialised high-cost equipment, not easily accessible to the average user.

## 2.3 Positioning hardware

The accuracy, quantity, and distribution of control measurements have a direct impact on georeferencing quality, and the accuracy of topographic products derived from SfM-MVS processing techniques. Georeferencing can be achieved either indirectly using GNSS measurements of ground control points (GCPs), or with direct measurements of camera positions synchronized

with image capture. The indirect (ground-based) approach requires an appropriate number, ideally over 20 (Carbonneau and Dietrich, 2017), well-distributed targets or distinct surface features which are then manually aligned on the captured images that they are visible in. While GCPs provide precise coordinates, the acquisition and processing of these points can be the

most time-consuming steps in the workflow, and it is impractical to have many (or even any) in large and difficult to access survey areas. The direct georeferencing method, using airborne control measurements, represents a major logistical advantage for aerial surveys in remote locations as it eliminates the need for a network of GCPs and, where camera position information is acquired with multi-frequency survey-grade GNSS equipment, it has been shown to produce results of similar precision to the ground-based approach (Nolan et al., 2015). Low cost single frequency GNSS receivers such as those onboard remotely piloted aircraft generate imprecise positioning solutions, accurate only to several metres. UAV-based aerial surveys relying on integrated GNSS sensors therefore often depend on GCPs for positioning (Carbonneau and Dietrich, 2017). The precision and accuracy of GNSS measurements can be improved using differential positioning, namely real time kinematic (RTK) or post processed kinematic (PPK), but this requires a direct communication link to a fixed base station which restricts the survey area to a limited operating range, and is impractical over large remote glaciers.

Kinematic positioning is more vulnerable to disturbances than static GNSS measurements and can produce highly variable results particularly where GNSS performance is less than ideal. At high latitudes, GNSS system performance is limited by satellite availability, receiver-satellite geometry, as well as increased ionospheric activity (Leick et al., 2015; Hugentobler and Montenbruck, 2017; Langley et al., 2017). The following section reviews the main factors influencing positioning quality from kinematic GNSS measurements onboard crewed aircraft, namely the performance of the GNSS hardware and its synchronisation with the camera.

### 2.3.1   GNSS receiver and antenna

GNSS positioning requires observations from a minimum of four satellites to determine pseudoranges and calculate receiver position and clock offset, while a fifth satellite is needed to solve for the tropospheric zenith path delay. The geometric dilution of precision (GDOP) describes the effects of satellite availability and configuration on the overall quality of the solution, with wider spacing between satellites resulting in more accurate positioning (lower GDOP) and tighter, or in-line, geometries degrading positioning accuracy (higher GDOP) (Langley et al., 2017). Satellite geometry is primarily limited by orbital inclination, which dictates the maximum satellite elevation above the horizon at a given latitude. The slightly higher orbital inclination of GLONASS satellites (65º) with respect to GPS, Galileo, and BeiDou (55–56º) is an advantage in polar regions, where GLONASS satellites reach maximum elevations of ∼55º above the horizon, compared to ∼45º for the remaining constellations. Overall, low satellite elevations weaken receiver–satellite geometry resulting in lower vertical position accuracy (higher VDOP). On the other hand, this situation increases satellite visibility in terms of numbers, as more orbital planes are visible at once, which improves horizontal positioning (lower HDOP) (Langley et al., 2017). Multi-constellation GNSS systems therefore benefit from increased satellite availability, which improves solution continuity and accuracy, and ensures stronger satellite geometry, even more so considering the higher orbital inclination of the GLONASS system. This is particularly valuable for kinematic applications where unpredictable motion can cause tracking issues and signal interruptions (Leick et al., 2015).

Signal propagation is affected by atmospheric (tropospheric and ionospheric) propagation delays, which are amplified for low-elevation satellite signals with longer transmission paths through the atmosphere. First, signal attenuation due to trans-

mission path loss is worse with low-incidence angles, which directly impacts the signal to noise ratio of the code and carrier phase measurements (Kouba et al., 2017; Langley et al., 2017). Second, increased ionospheric activity in the polar regions amplifies and causes more variability in atmospheric delays. As the magnitude of the delay is dependent on signal frequency, multi-frequency observations (such as L1, L2, and L5 for GPS) allow for correction of ionospheric delays (Kouba et al., 2017). GNSS signals at high latitudes are also affected by ionospheric scintillation responsible for causing irregular variability in signal phase and amplitude along the transmission path. Discontinuities in phase measurements can result in cycle slips and jumps in the carrier phase ambiguity, resulting in positioning errors of several metres or complete loss of lock. Multi-frequency observations are also used for cycle clip detection which allows the elimination of outliers and minimisation of positioning errors (Kouba et al., 2017). Multi-frequency receivers using the higher frequency L5 (GPS) band (or equivalent frequencies from other constellations) have an additional advantage in that higher frequency signals are less affected by ionospheric effects. This results in improved signal strength due to lower overall transmission path loss (Leick et al., 2015).

Multipath effects from surface reflectors are one of the main sources of error in GNSS positioning. Errors occur when reflections from multiple sources reach the antenna along with the direct line of sight signal (Leick et al., 2015; Langley et al., 2017). GNSS antennas for geodetic applications are designed to mitigate multipath effects using, for example, variable gain patterns to attenuate low-elevation signals and maximise signal strength at zenith, and different ground plane designs (e.g., choke ring, resistive 'stealth' ground plane) to improve multipath rejection blocking signals arriving from below the horizon (Leick et al., 2015). Onboard aircraft, satellite signal reception is additionally susceptible to airframe shadowing with the fuselage obstructing the line of sight. One solution is to position the antenna outside the aircraft, but signal dropouts can occur even in open sky conditions, either from the interference caused by helicopter rotor blades, or from disruptions due to aircraft manoeuvring. As the impacts of atmospheric propagation delays, multipath effects, and signal shadowing are particularly problematic with low signal incidence angles, satellites close to the horizon (15º or less) are usually ignored (Leick et al., 2015). However, this can be detrimental to precise positioning if satellite availability is low.

## 3   Field data acquisition

While knowledge of the theoretical concepts related to photography and GNSS navigation discussed in the previous section is important when considering hardware selection, understanding the limitations of both imaging and positioning systems is essential for developing data acquisition strategies in the field. Together with the general configuration of the survey site and type of aircraft used for the study, the imaging hardware determines image network geometry and ultimately the maximum achievable resolution and scale of the topographic reconstruction. The performance of the GNSS system is in turn responsible for limiting the precision and accuracy of control measurements, directly impacting georeferencing quality. Finally, depending on the camera used, image capture settings are chosen to ensure appropriate exposure given local light conditions, surface properties, and aircraft motion.

In this section we describe aspects of field data acquisition, including general survey design, control measurements, and image capture settings. To illustrate the practical realities of undertaking surveys in challenging conditions for photogrammetry,

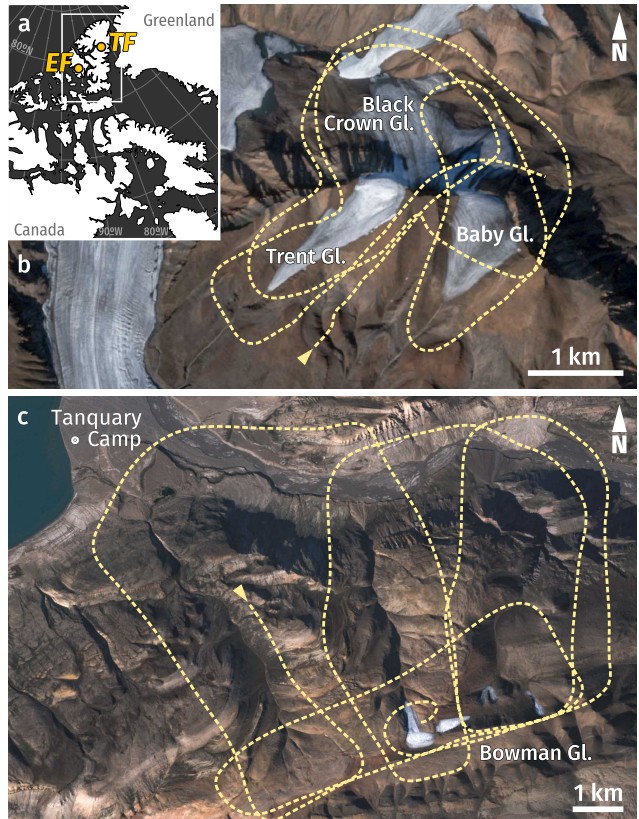

**Figure 5.** (a) Location of surveyed sites on Axel Heiberg and Ellesmere islands, Nunavut, Canada, and flightpath of air photo surveys over (b) Adams Icefield, Expedition Fiord (EF), Axel Heiberg Island (10 July 2019) and, (c) Bowman Glacier, Tanquary Fiord (TF), Ellesmere Island (03 Aug. 2018). The camera viewing direction was roughly orthogonal to the direction of travel (indicated by an arrow), looking left out of the aircraft in (b), and right in (c). Base images: Sentinel 2A, July 2020.

we particularly focus on settings used at two high latitude glaciers with target areas spread over large elevation ranges with steep ice and narrow valleys. Varying surface properties with a mix of snow, ice, and bare rock over rugged topography result in highly variable light conditions, changing between areas of bright snow and dark shadows.

The first study area is Bowman Glacier (81.35º N, 76.45º W; Fig. 5c), a small mountain glacier ∼10 km southeast of the Parks Canada basecamp at the head of Tanquary Fiord on northern Ellesmere Island, Nunavut. The glacier covers an area of <1 km² at an elevation range between ∼900 and ∼1200 m above sea level (a.s.l.). The survey at Tanquary Fiord (henceforth referred to as the TF survey) was flown on 3 August 2018 onboard an Airbus ASTAR 350 B2 helicopter. We covered 85 km of flightlines during an approximate flight time of 50 minutes, surveying a ∼70 km² area around Bowman Glacier and the main three valleys in its drainage basin (Fig. 5c). Due to logistical constraints, the survey took place around 2100–2200 local time, when the sun was quite low on the horizon, which resulted in long shadows and shooting images directly into the sun at times.

The second site is Adams Icefield (79.44º N, 90.59º W; Fig. 5b), a small icefield in Expedition Fiord, on the western coast of Axel Heiberg Island, Nunavut. Covering an area of ~3 km$^2$ at an elevation of ~700–1200 m a.s.l., the icefield is composed of three mountain glaciers (Baby, Trent, and Black Crown), which flow down either side of a ridge formed by three peaks, each reaching 1100–1250 m a.s.l. The Expedition Fiord survey (henceforth EF survey) was flown on 10 July 2019 onboard a Bell 206L Long Ranger helicopter, covered a considerably smaller area of ~10 km$^2$, and was surveyed in 20 minutes over 30 km of flightlines (Fig. 5b). Although the flight occurred around noon local time, one of the glaciers is north-facing and was therefore partially in shadow. Isolated patches of low clouds moving across the glaciers introduced an additional difficulty.

## 3.1 Survey setup

Due to the extensive area and complex topography of the two surveyed sites, images were acquired in a convergent geometry (roughly 30–50º off-nadir), along irregular flight paths around the area of interest. This differs from the more common geometry for airborne photo surveys, using nadir imagery captured in a regular grid pattern of parallel lines with a relatively constant aircraft velocity and height above ground surface, which ensures constant overlap between images and complete coverage of the area of interest. In mostly flat topography, a nadir pointing camera ensures an approximately constant GSD across all images, giving a consistent resolution and expected accuracy of the topographic reconstruction (e.g., Eltner et al., 2016; Smith et al., 2016a). Using oblique imagery in combination with a nadir dataset has been shown to strengthen image geometry and minimise systematic errors due to doming effects common to vertical datasets (James and Robson, 2014; Nesbit and Hugenholtz, 2019; Sanz-Ablanedo et al., 2020), with a convergent image geometry with varying angles oriented around a central area of interest showing the biggest improvement (Sanz-Ablanedo et al., 2020). An advantage of oblique image capture is an extended footprint, giving the ability to cover more ground in a single flightline, therefore reducing flying distance, time, and cost. Given the large size, complexity, and irregular relief of the two survey areas in this study, flying in a regular grid was impractical, while changing wind patterns resulted in variable flight speed, and large terrain elevation ranges caused the height above the ground surface to vary considerably (Table 2). Using nadir imagery, full coverage of the survey areas would have required a significant increase in flying height and/or shorter image capture interval. A convergent image geometry was also more suitable to survey the steep slopes and near-vertical rockfaces at both study sites, where oblique viewing angles were ultimately more orthogonal to the ground surface. In both surveys, the camera sensor was oriented with the short edge parallel to the direction of aircraft travel (yaw 0º), and the viewing direction roughly orthogonal to the flightpath (pitch 0º), varying between 30–50º off-nadir to the right at TF (roll >0º) and to the left at EF (roll <0º).

At TF we aimed to fly along each side of the three drainage valleys and around the summit plateau, to cover all features from multiple perspectives (Fig. 5c). The aircraft remained at a near-constant altitude of ~1400 m a.s.l., or between 50 m and 1350 m a.g.l. With the camera handheld, images were captured looking to the right, out of the open back door of the helicopter, with the camera operator tethered and leaning out of the aircraft to avoid catching the skid at the bottom of each shot. With no intervalometer available at the time of the survey, image capture was triggered manually at 2–5 s intervals depending on aircraft altitude above ground to maintain ~80 % overlap between consecutive photos, for a total of 1096 images (Table 2). With the glacier being the main focus, we aimed to maximise resolution over the glaciated summit plateau by flying close to the surface

**Table 2.** Details for the two survey flights in this study. The ground sampling distance (GSD), based on the indicated aircraft altitude a.g.l., represents an upper-bound estimate assuming nadir imagery. Lever arm refers to the physical offset between the GNSS antenna and camera sensor onboard the aircraft.

|  | Tanquary Fiord (TF) 2018 | Expedition Fiord (EF) 2019 |
|---|---|---|
| Date | 3 August 2018 | 10 July 2019 |
| Time (local) | 21:30–22:20 (50 min) | 11:30–11:50 (20 min) |
| Area surveyed | 70 km$^2$ | 10 km$^2$ |
| Flight distance | 85 km | 30 km |
| Altitude (a.s.l.) | 1200–1400 m (mean 1400 m) | 950–1350 m (mean 1250 m) |
| Altitude (a.g.l.) | 50–1350 m (mean 500 m) | 75–600 m (mean 350 m) |
| GSD | 10–245 mm (mean 90 mm) | 15–110 mm (mean 65 mm) |
| Velocity | mean 30 m s$^{-1}$ (max 40 m s$^{-1}$) | mean 27 m s$^{-1}$ (max 41 m s$^{-1}$) |
| GNSS logging rate | 15 s | 0.1 s (10 Hz) |
| Length of lever arm | 2.40 ± 0.43 m | 0.36 ± 0.17 m |
| Trigger intervals | Manual trigger, 2–5 s | Intervalometer, 2 s |
| Image count: total | 1096 | 551 |
| count: used | 980 (removed 116: blur) | 549 (removed 2: low tiepoint count) |
| count: w/ valid pos. | 851 (87 %) | 522 (95 %) |

(with short image capture intervals providing more data), and opted for a coarser GSD (longer image capture intervals) over the remainder of the drainage basin. This was an opportunistic choice rather than a deliberate strategy, and in general using an intervalometer is recommended to ensure proper overlap and avoid data gaps.

At EF we focused on flying around the three glaciers following their margins (Fig. 5b). Due to their small size, a single pass along each glacier margin was enough to cover the full ice extent. Here, we aimed to maintain a relatively constant height above the ground surface and fly following the topography, but in reality images were captured between 75–600 m a.g.l. In contrast to the TF survey, the helicopter back door could not be opened, and images were therefore captured looking left, out of an open window on the front passenger side. Image capture was triggered automatically with a wired intervalometer at regular 2 s intervals, yielding a total of 551 images (Table 2).

### 3.2  Control measurements

### 3.2.1  Camera positions

For both surveys, the GNSS system was set to record satellite observations at a 10 Hz logging rate, but due to a system malfunction at the start of the TF survey, the receiver only recorded observations at 15 s intervals throughout the flight. The GNSS antenna was positioned inside the aircraft. While an external mount would have been preferable for satellite visibility,

mounting any equipment on the outside of a chartered aircraft in Canada is not possible without prior regulatory approval. To minimise the negative impact of airframe shadowing on satellite signal reception, the antenna was held on the passenger (left)

side of the cockpit, where the larger windows offered better line-of-sight and improved satellite reception compared to the rear of the aircraft. One downside of this setup is the length of the lever arm, or the physical offset between the antenna (the recorded positions) and the camera sensor (the final positions to be estimated). Variations in the orientation of the lever arm caused by aircraft motion and attitude changes must be compensated for to determine absolute camera positions and minimise positioning errors. At EF, with both the GNSS antenna and camera positioned on the left side of the cockpit, the measured lever

arm was <0.5 m. At TF, with the camera located on the opposite side of the aircraft from the antenna, the estimated offset was 2–3 m.

    In this study, GNSS measurements were acquired using a survey-grade dual-frequency GPS (US satellite constellation only, L1/L2 bands) system consisting of the Trimble R7 receiver and Zephyr 1 Geodetic antenna. Following the strategy of Nolan et al. (2015), we acquired direct measurements of camera positions by synchronising the GPS receiver with the camera shutter

via wired connection using the Trimble Event Input Marker device. We additionally recorded the position of several targets on the ground around both surveyed sites, to serve as checkpoints for the direct georeferencing method. We use undifferenced satellite observations from a single GNSS system processed with precise point positioning (PPP). In contrast to differential positioning (i.e., RTK and PPK), PPP requires no fixed reference station and uses precise satellite orbit and clock products to correct for biases in estimated coordinates (Kouba and Héroux, 2001; Kouba et al., 2017). Removing the need for a second

GNSS system on the ground therefore reduces equipment costs and provides greater operational flexibility in remote areas.

### 3.2.2   Ground targets

To provide independent checkpoints to assess the quality of the direct georeferencing method (using camera positions), we recorded the position of two checkered flags (each $1 \times 1$ m in size) placed around both survey sites. The position of each point was measured with a Trimble R7 receiver within a few hours of the air photo survey, with a minimum occupation time of 20

minutes to ensure centimetre positional accuracy. At EF, we additionally measured the position of three distinct large boulders, on three separate occasions between one and two weeks after the survey. As these targets were only used as validation points for the direct georeferencing method, as opposed to proper control points, only a few were collected.

### 3.3   Image capture

Taking into consideration the main elements of an imaging system reviewed in Sect. 2.2, the camera, the lens, and image

capture settings, were chosen to optimise sensitivity and resolution and maximise the quality of image data recorded. All images were captured with a Nikon D850 camera and NIKKOR AF-S 24mm $f/1.8$G ED lens in Nikon RAW format (NEF), in aperture priority mode, thus keeping a constant aperture and letting the camera automatically adjust the shutter speed and ISO according to changing light conditions, ensuring proper exposure in both highlights and shadows. A common strategy used to maximise the signal to noise ratio in images is to expose to the right (ETTR), to essentially shift the histogram as far as possible

to the right (i.e., maximise brightness) without clipping the highlights. In most cases, this allows optimisation of exposure and

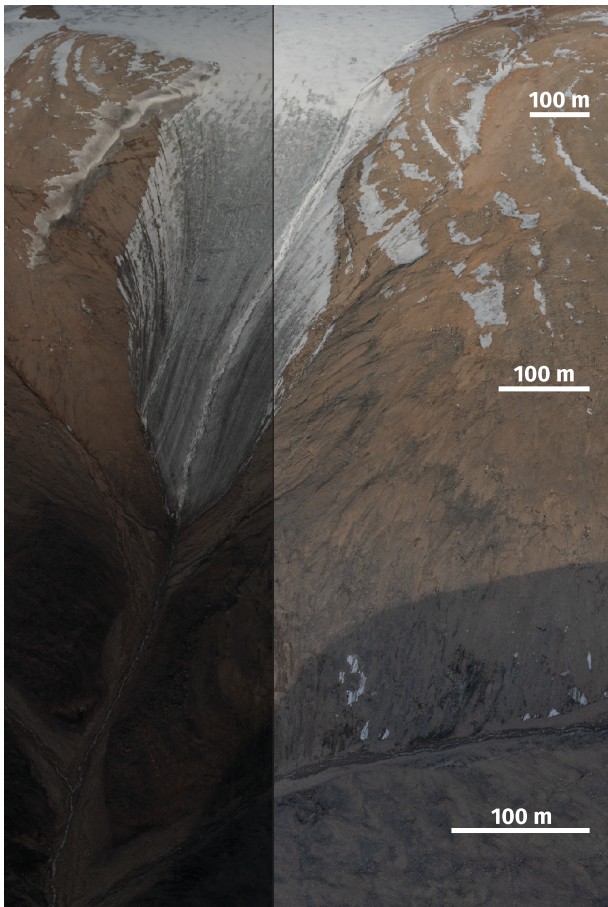

**Figure 6.** Strong shadows on an image of Bowman Glacier (TF) acquired on 3 August 2018. (Left) Unedited RAW image and (Right) edited copy with exposure adjustments to bring out information from the underexposed part of the image. As the distance between the camera sensor and ground surface increases towards the top of the frame, image scale in the foreground is about twice the scale in the background.

preservation of a maximum level of detail in the shadows (Rowlands, 2020, chap. 5, p. 70). However, during airborne surveys where on-the-fly exposure adjustments are impractical or entirely impossible, the safer approach (used here) is to expose to the left (ETTL) as it is less likely to overexpose highlights and lose crucial information over bright ice and snow-covered areas. The idea is to slightly underexpose the scene and bring out the shadows in postprocessing, at the expense of introducing more noise to the image (Rowlands, 2020, chap. 5, pp. 70–1). This can be done by setting the exposure value (EV) compensation to negative values. For example at TF, since the sun was low over the horizon and the light was also getting low, exposure compensation was set to –1 EV which forced the camera to underexpose the scene by a full stop (Fig. 6). This also increases the shutter speed and lowers the risk of introducing motion blur, while also decreasing ISO which is beneficial for lower noise and preserving distinct boundaries between features.

For both surveys, the aperture was set $\sim$3 stops under the maximum lens aperture of $f/1.8$ to minimise both diffraction effects (worse at apertures of $f/8$ and beyond; Fig. 2) and lens aberrations (amplified at large apertures; Sect. 2.2.2). At TF, with an aperture of $f/5.6$, shutter speed was limited to a minimum of 1/1000 s. Where there was insufficient light to maintain proper exposure, the camera automatically increased the gain from the minimum native ISO of 64, up to a maximum of ISO 800 which, for the D850, still produces acceptable levels of noise. For the EF survey, aperture was lowered to $f/5$ to prioritise faster shutter speeds (minimum 1/2000 s) while also lowering the maximum ISO (maximum ISO 400). Images were underexposed by 2/3 of a stop (–0.7 EV).

The minimum shutter speed was selected to avoid camera shake and ensure aircraft motion did not introduce additional blur, degrading image quality. With an average flying height of 350 m a.g.l., taking into account diffraction effects at an aperture of $f/5$, the diameter of the circle of confusion calculated from Eq. 2 is 98 mm, the equivalent of 1.5 times the GSD of 63 mm from Eq. 1. The amount of additional blur $b$ (in pixels) due to aircraft motion depends on travel velocity $u$ and shutter speed $t$:

$$b = \frac{u\,t}{GSD} \tag{4}$$

With an average aircraft velocity of 30 m s$^{-1}$ and minimum shutter speed of 1/2000 s, the circle of confusion is only 0.24 pixels (or 15 mm on the ground). In this case, the shutter speed is sufficiently high to avoid motion blur, meaning that diffraction effects (and other lens aberrations) would be primarily responsible for reducing the system resolving power. A certain amount of blur is expected and should be taken into consideration when devising the flight plan. Considering the highly variable flying height above ground for surveys over rugged terrain, and the resulting differences in GSD, the minimum shutter speed (and target aircraft speed) should be selected in order to keep the circle of confusion due to motion blur at or below the diameter of the circle resulting from diffraction (Eq. 2).

Lastly, achieving and maintaining proper focus is crucial. Here, the focus was set to infinity (by focusing on a distant feature) and switched to manual as conditions during a typical air photo survey make it difficult to ensure the proper functioning of the auto focus, which tends to pulsate in and out of focus between exposures. Unfortunately, about 10 % of the TF images ended up slightly out of focus (and were discarded) due to the camera operator accidentally nicking the focus ring 20 minutes into the survey; taping the focus ring is a common strategy (and generally a good idea) often used to avoid this issue.

## 4 Raw data processing

### 4.1 Image postprocessing

Following the surveys, the raw image data were first processed to maximise the amount of visual information in order to help with feature extraction. Initial corrections were performed on the 14-bit RAW images in Adobe Lightroom (Process Version 5) using the lens profile listed in the EXIF metadata to correct for vignetting and remove chromatic aberrations, while leaving any geometric distortions untouched. Chromatic aberration is caused by light refraction which causes colour fringing, or purple artefacts along high-contrast boundaries (Fig. 7). Vignetting consists of illumination falloff, a drop in light intensity towards the edges where light reaches the sensor at an angle (Ray, 2002, pp. 132–3). Since photogrammetry algorithms rely on

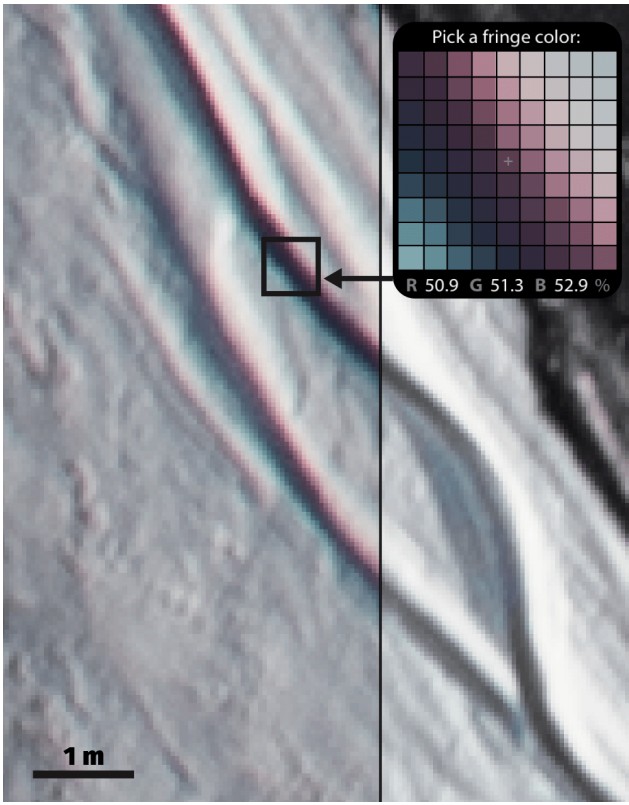

**Figure 7.** (Left) Uncorrected image from the Expedition Fiord (EF) survey (10 July 2019) showing chromatic aberrations with purple fringe along meltwater stream. (Right) Corrected image with aberrations removed.

distinct edges between features and consistent lighting conditions, compensating for both these aberrations should theoretically enhance feature detection.

The digitisation process which converts sensor information into raw image data introduces some softness, which was compensated for by applying the default sharpening presets in Lightroom. Although additional sharpening can improve perceived sharpness, especially where image quality is lacking, excessive sharpening can also impact the structure of the file and introduce visible artefacts, such as halos at object edges (Rowlands, 2020, chap. 5, p. 40). Additional digital noise reduction was performed to minimise chromatic noise (colour variations between pixels in areas of uniform colour, mostly apparent in dark shadows), which becomes more problematic with increasing ISO values. Lastly, a variable exposure gain was applied to all images to brighten underexposed areas and increase the level of detail and available information for feature extraction. Here, Lightroom automatically adjusts the total exposure (EV) of successive images captured with different in-camera exposure settings (i.e., shutter speed and ISO), to match a selected reference image. This was performed in batches, selecting overlapping images with similar content to that of the reference image, to even out differences in illumination between images and enable a more uniform orthophoto reconstruction. Images were ultimately exported as uncompressed TIFFs with 16-bit depth, yielding

files of >270 MB each (total dataset size of over 400 GB for the 1529 images from both surveys, excluding the discarded and out of focus ones).

Both surveys were flown in a convergent geometry and therefore included few nadir images, with most being taken at a low or high oblique angle. For high oblique photos, where the horizon is included in the frame, the most time-intensive task in post-production is masking extensive swaths of sky and any terrain beyond the area of interest. A few images also included

some part of the helicopter skid. For EF with a high proportion of high oblique images, 90 % of all images required masking, compared to 17 % for TF where images were captured at a low oblique angle.

## 4.2 GNSS data postprocessing

Camera positions associated with each captured image were calculated from PPP processed GNSS measurements. The raw (GPS) satellite observables collected by the dual-frequency (L1 and L2) Trimble R7 receiver were first converted into RINEX

format with the Trimble Convert to RINEX Utility and then processed in kinematic mode (i.e., with a unique position for each epoch) using the Geodetic Survey Division of Natural Resources Canada Canadian Spatial Reference System PPP (NR-Can CSRS-PPP) online service (software version 3.45.0, updated 27.10.2020, https://webapp.geod.nrcan.gc.ca/geod/tools-outils/ppp.php). The steps for retrieving camera positions at the time of image capture from this data were implemented in a custom R (version 4.0.5) script and are summarised below. The uncertainty of camera positions was determined with a series

of estimations to take into account various error sources, including GNSS positioning and PPP modelling errors, imprecise lever arm measurements, unmodelled aircraft attitude changes, and delays in synchronisation between the positioning and imaging systems.

### 4.2.1 PPP processing

The CSRS-PPP positioning tool takes into account observations from all satellites in view above an elevation mask angle of 7.5º

and requires a minimum of five satellites for a valid position fix. The solution is derived from undifferenced code and carrier phase observations from a single GNSS receiver and relies on precise satellite orbits and clock products from the International GNSS Service (IGS). It uses a sequential Kalman filter to estimate four unknown parameters: receiver position, receiver clock offset, tropospheric zenith delay, and carrier phase ambiguities. In kinematic mode, the algorithm solves for independent solutions at each observation epoch and uses backward smoothing to produce a corrected track (Kouba and Héroux, 2001).

The PPP output includes the position solution, as well as the predicted level of uncertainty given as 2-sigma error bounds (95 % confidence level).

### 4.2.2 Camera positions

Synchronisation between the camera shutter and positioning system was achieved via the Trimble Event Input Marker connecting the GPS receiver to the camera hot shoe accessory mount. With each shutter action, the camera generates a TTL

(Transistor-Transistor-Logic) level voltage signal which is generally used to activate an electronic flash unit. Here, the rising

edge of the TTL pulse output by the camera was recorded by the GPS receiver as an external event, which is listed as an event flag record alongside the satellite observation records in the RINEX observation file. The PPP software estimates coordinates for epochs with valid satellite observation records but skips over all event flag records. Therefore, the position of the antenna at the precise moment of image capture has to be subsequently interpolated from the estimated GPS positions immediately

before and after each event record. Corresponding camera positions are then calculated, compensating for the physical offset (lever arm) between the GPS antenna and camera sensor. The steps used for assigning coordinates at each event and calculating camera positions at the time of exposure are described in detail in Medrzycka (2022).

### 4.2.3 Synchronisation error

The synchronisation between the positioning system and camera shutter depends on the sum of delays due to the GNSS receiver

clock, length of cable runs, and camera operations. As discussed in Sect. 2.2.4, focal plane shutters (such as the one on the D850) use mechanical vertical travel shutters which expose the sensor array sequentially over a timespan corresponding to the flash synchronisation (X-sync) speed (1/250 s for the D850) after which the shutter is fully open, and the TTL signal is generated. Considering the propagation delay and rise and fall time for a standard TTL signal and the typical accuracy of the timestamp logged by the GPS receiver, in theory the synchronisation between camera shutter and the GPS is $<1\,\mathrm{ms}$. However,

when capturing images from a moving platform, taking into account the vertical curtain travel time and the resulting motion trace significantly increases synchronisation uncertainty (by 3 orders of magnitude). With an average travel speed of $\sim30\,\mathrm{ms}^{-1}$ ($\sim60$ knots), combining the X-sync speed (1/250 s) with the slowest shutter speed used (EF 1/2000 s; TF 1/1000 s) means that, on average, the camera travelled $\sim12$–14 cm during a single exposure. The computed horizontal and vertical components of the motion trace calculated for each image are therefore taken into account in the final camera position uncertainties. Camera

positions interpolated over data gaps with no GNSS solution were marked as invalid and therefore not used in the following SfM processing steps.

## 5 DEM & orthophoto generation

Following postprocessing of raw field data, optimised image files and corresponding camera position estimates were used to derive georeferenced DEMs and RGB orthomosaics of the study areas. The full SfM-MVS workflow was performed in

Agisoft Metashape Pro (version 1.6), starting with feature detection and SfM processing, followed by MVS matching and dense cloud generation. Additional point cloud filtering and the final error assessment were undertaken in CloudCompare (version 2.11.1, cloudcompare.org). Given the focus of this paper on optimising survey design for challenging field conditions, we don't describe the full SfM-MVS processing here in detail (full details are provided in Medrzycka (2022)), but instead summarize the main outputs (Fig. 8) and discuss their relevance for field surveys.

In both surveys in this study, flightlines were deliberately planned to prioritise image acquisition over ice masses and areas of lesser interest received lower coverage or were only imaged from afar. As a result, the quality of the reconstructions varies spatially and is most consistent over glaciers and adjacent areas. Where coverage is adequate and the surface uniform and fairly

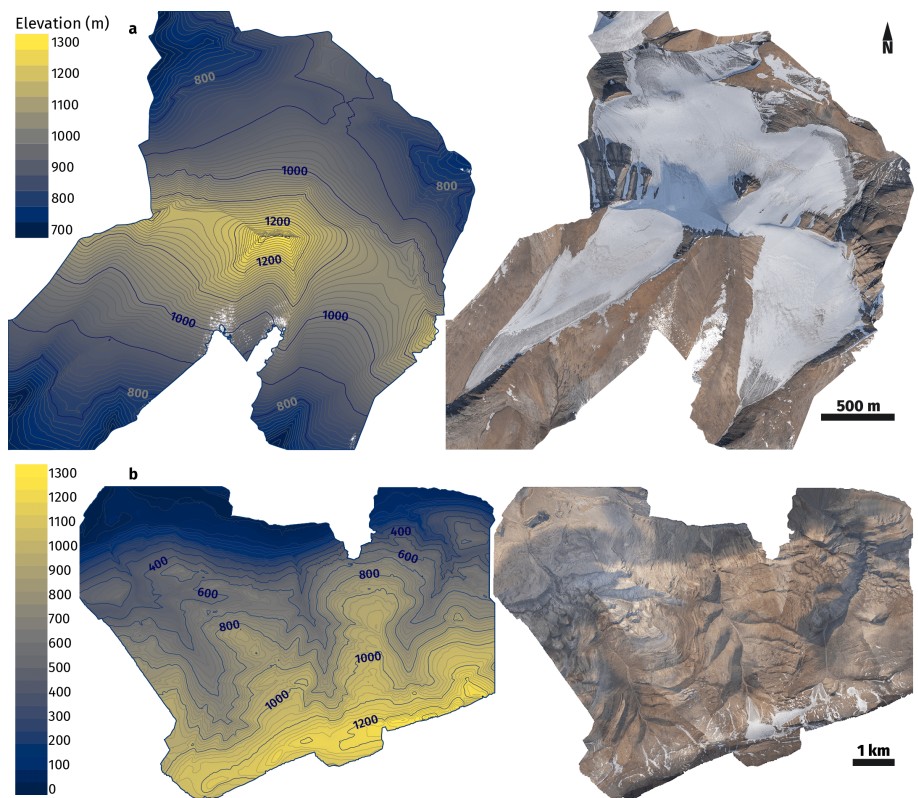

**Figure 8.** SfM-MVS outputs with elevation model (left) and RGB orthomosaic (right) of the (a) Expedition (EF) and (b) Tanquary Fiord (TF) survey areas. Elevation contour lines at 10 m spacing in (a) and 20 m in (b). Same spatial coverage as in Fig. 9.

flat, point spacing in the final reconstruction is regular and surface density more or less constant, with an average of $13.75 \pm 1$ pt m$^{-2}$ and $21.5 \pm 0.8$ pt m$^{-2}$ at TF and EF, respectively (Fig. 9). Overall, the main difference between the two surveys is likely related to aircraft altitude above ground, which was on average higher at TF (500 m) than at EF (350 m). For a given imaging system (with a given resolving power and focal length), the distance from camera sensor to ground surface is directly related to the size of pixels on the ground (the GSD). Objects imaged at close range cover more pixels on the sensor and are relatively larger and so more detailed in image space. Higher information content in image data enhances feature detection (both quantity and quality) which in turn determines the maximum achievable point density. Low altitude surveys therefore allow for higher resolution surface reconstructions but are impractical over large study sites and those with highly variable surface topography. Steep terrain inevitably forces an aircraft to increase flying height in order to steer clear of peaks and ridges, meaning that low lying points are mostly imaged from further away. At the same time, image footprint increases with distance to object, which provides wider spatial coverage in a single pass, reducing both flying distance and survey duration.

In addition, there is considerable spatial variability in point density within each survey. At TF the reconstruction tends to be patchier between each of the three drainage valleys (Fig. 9b). Due to oblique image capture, the camera was able to image a

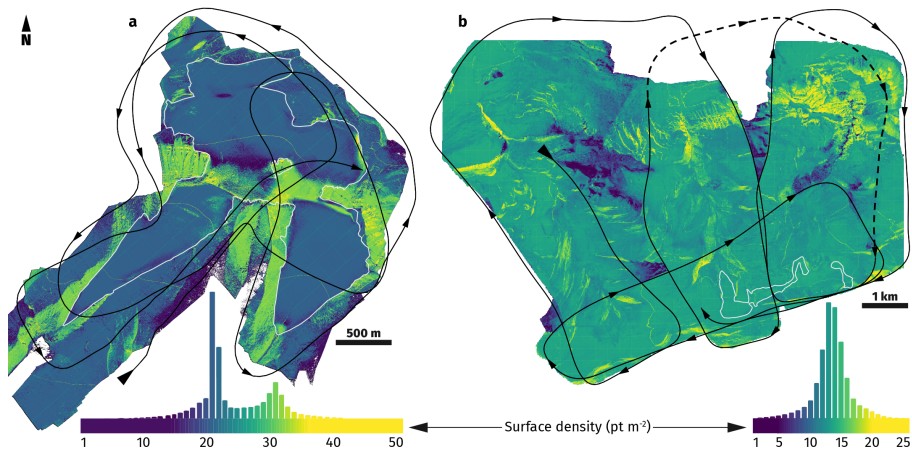

**Figure 9.** Surface point density for the (a) Expedition (EF) and (b) Tanquary Fiord (TF) dense point clouds. White outline indicates glacier ice. Solid black line shows the flightpath and direction of aircraft travel. The dashed line segment in (b) represents a 10 km segment where 116 out of focus images were removed from the dataset. Same spatial coverage as in Fig. 8.

single side at a time and any terrain on the opposite side, or immediately below, the aircraft was either omitted entirely or only appeared in the background of a few images. At EF, steep slopes ($>$30º) tend to have higher surface point density, averaging $31.5 \pm 1.4 \, \mathrm{pt \, m^{-2}}$ (Fig. 9a). Unlike the more spread out TF survey site, the EF survey was focused around a central massif which, combined with lower aircraft flying height, favoured more oblique image capture. As a result, steep vertical terrain

was imaged at a more perpendicular angle, which helped minimise perspective distortions between overlapping images and allowed for more reliable extraction and consistent identification of distinct surface features in stereo matching. In Metashape, feature matching is based on the scale-invariant feature transformation (SIFT) algorithm (Lowe, 1999) which, while efficient at matching features despite variations in scale and orientation, is only partially invariant to illumination and affine distortions (Lowe, 2004). In general, extreme affine distortions from widely different viewpoints such as those common in low-oblique

aerial imagery pose a significant challenge to robust feature matching. In addition, oblique viewing angles are more likely to cause occlusions, where distant objects are obscured by those in the foreground, further reducing matching performance (Lowe, 2004). Lower point density is also characteristic of deeply incised V-shaped valleys, as the valley bottom is often in shadows and easily obscured from view by steep slopes rising on each side.

In this study, the only source of independent validation are the checkpoints measured in the field. Georeferencing accuracy

varies spatially across the reconstruction, and so checkpoints are only representative of local registration errors. Still, the magnitude of these values is a useful indicator of the maximum expected accuracy when characterising surface topography. Here, RMS errors for the two checkpoints at the TF survey site are 0.35 m horizontal and 0.68 m vertical (0.77 m total), and for the five checkpoints at EF, 0.46 m horizontal and 0.29 m vertical (0.56 m total).

Ideally, horizontal accuracy would be higher or equal to the spatial resolution of the final gridded products which, for surveys

with more regular geometry and constant height above ground, should be roughly equivalent to the GSD. In this case, where

GSD is not easily constrained, point density is useful to define an appropriate pixel size for the gridded products. Here, DEMs and orthomosaics were gridded at 0.5 m, or roughly half the achievable resolution based on the average point spacing of 0.27 m at TF, and 0.22 m at EF. The 0.5 m represents a compromise between processing time and resolution and, in this case, is sufficiently detailed to answer the requirements of the specific project. Horizontal checkpoint misalignment errors remain below the 0.5 m cell size for both reconstructions. Vertical accuracy requirements are project specific and depend on the level of detail necessary for a given study, but are sufficient for detecting long-term changes in the geodetic mass balance of glaciers in this study given surface thinning that can exceed $>1$ m a$^{-1}$ at low elevations in the Canadian Arctic (Thomson et al., 2017).

## 6  Discussion & recommendations

Based on the surveys conducted at Expedition and Tanquary fjords and given our experience with hardware selection, field data acquisition strategies, postprocessing steps, and the generation of final DEM and RGB orthomosaics, it is useful to provide a critical evaluation of each aspect and develop a set of recommendations for improving aerial surveys to achieve optimal results in suboptimal conditions. These are detailed below.

### 6.1  Imaging system

#### 6.1.1  Camera and lens

The camera (Nikon D850) and lens (NIKKOR AF-S 24mm $f$/1.8G ED) used in this study were selected based on the key aspects discussed in Sect. 2.2, prioritising high resolving power, high SNR, and low geometric distortions. Low-cost compact cameras, including smartphones, action cameras, and cameras commonly mounted on consumer-grade RPAS, have been shown to provide adequate results for some applications, but comparisons with higher performance imaging systems show significant improvements in final outputs (e.g., Thoeni et al., 2014; Eltner and Schneider, 2015; Micheletti et al., 2015; Girod et al., 2017; Stark et al., 2021). With image resolution (GSD) being the main contributing factor to result accuracy, studies have also demonstrated the importance of maximising the level of detail captured (Mosbrucker et al., 2017), avoiding extreme distortions from very wide-angle (fisheye) lenses (Thoeni et al., 2014; Girod et al., 2017), minimising rolling-shutter effects with fast sensor readout speeds or global shutters (Vautherin et al., 2016; Zhou et al., 2020; Stark et al., 2021), and ensuring precise synchronisation between the camera and positioning system (Welty et al., 2013).

A key point here is that GSD, which is often stated as the only metric for spatial resolution, is only part of the equation. The GRD, from the contribution of system optics including diffraction effects, aberrations, motion blur, and general defocus, has to be taken into consideration prior to data acquisition in the field. Maximising image information content enhances feature detection and matching performance, resulting in higher point density, and significantly improving both reconstruction accuracy and precision (Gienko and Terry, 2014; Mosbrucker et al., 2017). In this study, feature detection was performed after upsampling all images by a factor of four. Increasing image resolution involves interpolating pixel values which can introduce artefacts (Rowlands, 2020, chap. 5, p. 44), and is therefore only advantageous for high resolution and high quality images

where it can be helpful with identifying more features and matching them with sub-pixel precision. Tested on a subset of 20 images from the EF dataset, the average number of features detected on upsampled images was 7–8 times greater than with the original resolution data (1.6 vs 0.2 million points per image), and the average number of valid matches increased by a factor of four (310,000 vs 75,000 points per image). The total point count in the corresponding clouds was five times greater based on the upsampled data (2.3 vs 0.46 million points each). This underlines the importance of selecting high performance imaging hardware maximising resolving power and image quality to ensure high point density, and in turn more accurate topographic reconstructions.

As for geometric distortions, high build quality components with stricter manufacturing requirements ensure a more stable internal camera geometry, including precise alignment and perpendicularity of the optical axis to the focal plane. For the camera body and lens, magnesium alloy offers greater stability than polycarbonates and lowers the risk of mechanical instabilities from thermal effects, vibrations, and shocks. Ideally, a camera should be selected with a global shutter to avoid rolling shutter effects and ensure precise synchronisation between the camera shutter and positioning system. Otherwise, a high X-sync speed (preferably 1/250 s) is necessary to (1) minimise the distance travelled by the camera during exposure and enable more precise camera positioning, and (2) reduce equivalent displacements of imaged objects in pixels to obtain lower distortions in resulting images. Minimising sensor readout time, as well as any deviations of the optical axis, will avoid having to compensate for affinity and non-orthogonality of image pixels and remove at least one unknown parameter from the camera calibration matrix in the alignment step. Additional key recommendations for imaging components include selecting:

– A high resolution camera with large sensor (ideally full-frame) for good sensitivity and spatial resolution.

– A high bit-depth (12–14 bits) sensor for high radiometric resolution and maximal image information in highlights and shadows.

– A prime lens (with a fixed focal length) to ensure stability of the internal camera geometry. In addition, prime lenses typically have higher quality optics with lower aberrations and geometric distortions compared to zoom lenses, which results in higher image quality.

– A wide angle lens with a focal length of 24–35 mm for a wide field of view, giving a large image footprint while minimising geometric distortions.

### 6.1.2 Image data capture

Capturing high quality image information requires full control over data acquisition, and therefore a camera with the option to control exposure and other in-camera settings during the survey. For more control and flexibility in postproduction, it is important to record images in RAW format to retain the full bit-depth as captured by the sensor and minimise the loss of information that occurs when using compressed file formats. Image capture and survey planning are tightly related to the size, topography, and surface characteristics of the study area. For a given camera and lens combination, flying height will determine the GSD (Eq. 1), which should be selected based on the project and required level of detail, while keeping in mind the extent of

the survey area and the flight time and number of images needed to provide full coverage with sufficient overlap. Other aspects should be configured to maximise the GRD while ensuring adequate exposure, keeping diffraction effects (Eq. 2) and motion blur (Eq. 4) to a minimum, ideally at or below 1.5 pixels. Maintaining image sharpness is crucial and, in low light conditions, decreasing flight velocity will allow faster shutter speeds while also reducing distortions from rolling shutter effects (Eq. 3). With large survey areas or limited flight time, where decreasing flight velocity is not possible, or with variable wind conditions causing sudden jumps in aircraft motion, it is preferable to increase ISO at the cost of introducing some noise, rather than risking blurry images.

Specific considerations concerning image capture settings and camera configuration can be summarised as follows:

- Avoid fully automatic shooting modes; ideally use aperture priority mode to avoid the internal lens configuration changing during the survey and affecting the camera calibration matrix.

- Use an aperture two to three stops from the maximum, ideally around $f/4.0$–5.6, to minimise diffraction effects and aberrations which degrade image quality.

- Use a high shutter speed (1/2000 s should be enough on most cameras) to avoid camera shake and keep motion blur to a minimum.

- If using aperture priority mode, set EV compensation to –0.3 EV or lower depending on light conditions in order to avoid high ISO and/or slow shutter speeds.

- If capturing JPEGs or other lossy formats, minimise in-camera processing, keeping picture mode 'flat' or 'neutral'.

- Select a wide colour space such as Adobe 1998 RGB instead of sRGB.

- Disable in-camera corrections which compensate for aberrations, especially geometric distortions.

- Disable image stabilisation and other options with the potential to modify image and/or pixel geometry.

- Use an automatic intervalometer for constant image capture intervals.

- Use a sturdy tripod and avoid resting any part of the camera against the body of the aircraft, which will transfer vibrations to the camera body and lens. Alternatively, with the camera handheld, the (camera operator's) body will dampen much of the vibration.

- Use manual focus and ensure that it stays in focus, such as by taping the focus ring.

- If possible, plan for best light conditions and no strong/long shadows. Overcast sky works best for uniform lighting. Avoid early morning or late evening when sun is at a low angle.

### 6.2 Control measurements

#### 6.2.1 Satellite availability

In this study, observations were limited to the 31 operational satellites in the GPS constellation, of which less than half were theoretically visible at any one time. Combining these observations with those from the three other major GNSS constellations would have doubled that number (from ∼15 to ∼30 potentially visible satellites; https://www.gnssplanning.com). Increased satellite availability can be especially important at high latitudes where poor GDOP due to low signal incidence angles, unfavourable receiver-satellite geometry, and higher ionospheric activity combine to degrade GNSS performance (Leick et al., 2015; Hugentobler and Montenbruck, 2017; Langley et al., 2017). For most of both surveys in this study, at least 10 satellites (maximum 12) were observed at any one time, however only in a few cases (<10 % for EF and <5 % for TF) were all of the observed satellites used in the PPP solution. For over half of all epochs, at least two satellites were excluded either due to a low elevation angle above the horizon (<7.5°), incomplete observations, or large signal residuals. Considering that a minimum of five satellites is required for a valid position fix, and that ∼20 % of all epochs in both surveys here either have the bare minimum required or are missing valid solutions entirely, increasing satellite availability would increase data continuity while also lowering the GDOP. As of yet, the CSRS-PPP online service is only set up to process GPS and GLONASS observations, but future improvements can be expected to ensure interoperability and compatibility between multiple GNSS constellations and signal frequencies.

#### 6.2.2 Data logging rate

To investigate the influence of logging rate on the error distribution for PPP solutions, we compared the original EF data, with satellite observables at 10 Hz intervals, to the same observations decimated to 15 s intervals. This shows the predicted uncertainty to be strongly dependent on sampling rate, with the 2-sigma position uncertainty estimates for both the decimated 15 s EF data and the original 15 s TF observations an order of magnitude higher (>1–3 m) than those for the 10 Hz EF dataset (<0.5 m). A higher sampling rate also helps smooth out some of the variability due to noisy measurements, and more frequent measurements improve data continuity. In case of signal loss, a single missing observation from the TF survey results in a 30 s data gap, and is therefore likely to be more disruptive than a few missing observations from the 10 Hz EF dataset.

Both lower logging rates and data gaps affect the performance of the PPP algorithm and are problematic for georeferencing since camera positions between two PPP estimated positions are determined using spline interpolation, with uncertainty growing over time with increasing distance from a position fix. Excluding data gaps, the average distance between any camera and the closest position fix is <1 m at EF, and ∼100 m at TF. Receiver position estimates between the original high-rate (10 Hz) and decimated (15 s) EF differ by ±1 m in Easting and Northing, while the height difference grows to nearly 4 m over the duration of the survey. However, where data gaps occur, the interpolated point coordinates differ by up to a few tens of metres in the horizontal. Due to data gaps in the GNSS observations, camera position estimates for 28 images (∼5 %) from the 10 Hz EF survey, and 129 images (or 13 %) from the 15 s TF survey, were marked as invalid and were omitted from the SfM workflow.

These observations on the impact of GNSS signal loss and resulting data gaps underline the benefits of improving positioning
data continuity. A key recommendation, particularly for surveys undertaken at high latitudes, is to use a multi-frequency and multi-constellation GNSS system, preferably one combining GPS, GLONASS, and Galileo observations. Although perhaps difficult to plan in remote field situations with limited aircraft availability, scheduling the survey during optimum satellite availability could help ensure lower GDOP for more precise measurements. Ensuring a high data logging rate ($<1$ s or better) is crucial for both data continuity and positioning precision. New GNSS receivers also offer higher sensitivity and can track
weaker incoming signals and lower measurement noise, enabling higher positioning accuracy. Likewise, newer generation GNSS antennas have higher gain for stronger signal receiving power and improved multipath suppression (Leick et al., 2015). This would be particularly useful in dynamic scenarios onboard moving platforms where frequent cycle slips and high multipath errors affect positioning accuracy. If possible, mounting the antenna outside the aircraft and maintaining shallow bank angles during turns would somewhat help mitigate potential signal loss from aircraft shadowing. Antenna placement should also be
considered in relation to the camera sensor, and the offset between the two should be minimised in order to reduce uncertainties in camera position estimates from aircraft attitude changes. Integrating the GNSS system with an inertial navigation system (INS) could have provided a significant improvement in positioning performance in this study by (1) recording attitude changes during flight to account for the effects of the lever arm, and (2) improving continuity by helping to approximate receiver positions in case of satellite signal loss (Farrell and Wendel, 2017).

**7    Conclusions**

In this study, we present theoretical concepts and practical considerations for the application of photogrammetry techniques for mapping glaciers in remote places, where aerial surveys are often performed in challenging conditions, including large study sites with complex topography, difficult light conditions, and poor GNSS navigation performance. Based on the results from two surveys conducted at Expedition and Tanquary fjords in the Canadian Arctic, we identify key limitations and propose a set
of recommendations to maximise the quality of results given imperfect conditions.

Together with the specific requirements and expected results of each study, the location, size, and topography of a survey site are the primary aspects determining general survey design. Hardware selection and data acquisition strategies in the field determine the quality and information content of raw data on which every other step in the processing workflow depends. Optimising results therefore requires flexible survey design and the use of high-performance imaging and positioning hardware
to enable the acquisition of large quantities of high-quality raw data, with the obvious trade off being increased cost and higher processing power requirements. In terms of image data, the main aspects are high resolving power, high SNR, and low geometric distortions, all of which are crucial for maximising visual content and ensuring robust feature matching over variable topography, and despite perspective distortions. Key considerations for optimising direct georeferencing include positioning data continuity, which primarily depends on GNSS satellite signal availability and integrity, and data logging rate.
Beyond any challenges inherent to the survey areas, logistical time constrains, and weather delays, the outcomes of the two surveys presented in this study were adversely impacted by unforeseen complications from system malfunctions (GNSS

receiver defaulting to low logging rate) and human error (forgetting the intervalometer, accidental loss of focus). However, the main weakness was in the positioning performance being limited by the GNSS system which, despite consisting of dual-frequency, survey-grade components, was only able to record GPS observations (i.e., from a single satellite constellation). With continuous developments in GNSS technology, and multi-constellation/frequency equipment becoming increasingly accessible, upgrading the positioning system used here will likely result in fewer data gaps and significantly improve direct georeferencing accuracy in future surveys. Additional INS data could further help derive more precise control measurements taking into account variations in aircraft attitude. When coupled with precise synchronisation with the camera sensor, INS-aided GNSS navigation could help reduce uncertainties in camera position estimates and ensure more accurate topographic reconstructions.

The accessibility of low-cost equipment and the largely automated SfM-MVS workflow requires little expertise in photogrammetry techniques, or the related fields of GNSS technology and satellite navigation, imaging optics, and signal processing. On the other hand, unlike many remotely sensed satellite products which come pre-processed and analysis-ready, SfM-MVS processing demands one to be more than an end-user. Integrating all aspects of the workflow is challenging and data quality can be highly variable from one study to the next. A significant challenge for undertaking a comprehensive error assessment is determining the exact contribution of all components involved and quantifying their effect on the final results. While the ultimate goal is mapping a given surface with high precision and accuracy, conducting surveys in suboptimal conditions where results are not necessarily expected to be of the highest quality, provides an opportunity for improving our understanding of those factors with the highest influence on final outputs and study outcomes.

When building long-term datasets, recording raw unprocessed data (avoiding lossy formats) is especially valuable for monitoring dynamic processes, including in glaciological studies where small ice masses are disappearing at increasing rates. Repeated surveys to monitor ongoing changes, and the collection of high-resolution data to allow continued investigations after those glaciers are gone, are invaluable for long term records.

*Author contributions.* DM, LC, LT, and BS contributed to survey planning and data acquisition in the field. DM processed raw field data, developed the code for calculating camera positions, and set up the SfM-MVS processing workflow. DM and WK carried out the processing and generated final outputs. This work is adapted from a chapter from the PhD thesis of DM, who prepared the manuscript with contributions from all co-authors.

*Competing interests.* The authors declare no competing interests.

*Acknowledgements.* This work was supported by Parks Canada, Natural Sciences and Engineering Research Council of Canada, ArcticNet Network of Centres of Excellence Canada, Polar Continental Shelf Program, Canada Foundation for Innovation, Ontario Research Fund, Ontario Graduate Scholarship and the University of Ottawa. We thank the Parks Canada staff at Tanquary Fiord, Quttinirpaaq National Park

for field support, and J. Rajewicz for assistance with fieldwork. We acknowledge the Nunavut Research Institute and the communities of Grise Fiord and Resolute Bay for providing permission to work at Tanquary and Expedition fjords.

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
