# Peer review of "Review of methodological considerations and recommendations for mapping remote glaciers from aerial photography surveys in suboptimal conditions"

_Geoscientific Instrumentation, Methods and Data Systems, 2024_

## Author Comment (AC1)

**Reply to RC1**

The paper presents many important aspects in order to achieve optimal results in photogrammetric surveys from the planning stage to the final results. These aspects are mostly reviewed from a theoretical point view. The paper ends with two practical examples presenting the work and outcome for two flights at high latitude (~ 80° North).

The paper is a bit of a mixed bag. The first part (approx. 11 pages) summarizes many aspects of modern day photogrammetry, including MTF, diffraction, motion blur, rolling shutter, GNSS specifics, etc. Actually, everything is true for any photogrammetric endeavor -- irrespective of the terrain type (glacier or not).

The 2nd part (also 11 pages) presents the work and outcome for two flights at high latitude (~ 80° North). Following the very detailed description in the first part, one might expect that the practical part then showcases how the theory of the first part is considered in the planning stage of the two flights, and/or that it is demonstrated that neglecting certain aspects from the theory section produces certain errors in the results. But that's not the case. Both flights appear to be executed without having had many options. E.g. what could have been investigated:
- different apertures
- effect of flight speed on blur
- shooting not in raw; or bad Adobe Lightroom settings when "developing" the raw images
- impact of these settings on the different surface types: glacier (which is also part of the title) and non-glacier
- impact of changing the oblique viewing angle
- different flight patterns
- etc.

Of course, the possibilities of changing the parameters and investigations are almost endless, but in the present form the photogrammetric results are obtained in <<one>> way. Apparently, the outcome fulfilled the requirements, but it remains a bit unclear if less strict settings would have let to similar acceptable results. Still, the paper is very well and understandable written. The theoretical part gives a great theoretical summary on many aspects. And the description of the two flights shows the reality of conducting photogrammetric data acquisitions in such high altitude.

We thank the reviewer for their useful comments, and provide detailed responses below. In relation to the comments about the general structure of the paper, it is true that our field measurements were not undertaken in a systematic way to investigate all of the available imaging options. Rather, as stated by the reviewer, the flights were executed without having had many options due to the remote nature of our fieldwork and limited time (particularly helicopter availability) to undertake our measurements. Our aim, therefore, is to demonstrate that real-life surveys can still produce useful results even when undertaken is suboptimal conditions, and for our paper to provide a bridge between purely theoretically-based studies and purely field-based studies, since few other papers have previously been published in this area.

In order to accept the paper I would ask the authors to provide a few more details in the theoretical and practical parts:

row 78 "Along with focal length, sensor size also defines the ground sampling distance (GSD) and therefore"
--> This is not correct, equ (1) for the GSD has no sensor size, just the pixel size (pixel pitch).
Thanks for point out this error. Changed to: "Along with focal length, sensor size and pixel count also define ..."

Table 1 "cy per in" should be "cy per mm".  Corrected "cy $\ln^{-1}$" to "cy $mm^{-1}$".
How is the Nyquist limit calculated? Added a sentence in the table caption:
"The Nyquist limit, defined in units of cycles per mm (cy $mm^{-1}$), is related to pixel pitch ($\rho$) by  $0.5/\rho$, and is equivalent to one half of the sensor cut-off frequency."

Fig. 1+2+3 The paper should include the formulae that went into creating these figures.
Added the following text to captions for:

Fig. 1 "Detector and AA filter MTFs are modelled with: $MTF_{det} = sin(\pi\nu\rho)/(\pi\nu\rho)$ and $MTF_{AA} = cos(\pi\nu\rho)$ (Rowlands, 2017)."

Fig. 2 "Lens MTF is modelled with: $MTF_{diff} = \frac{2}{\pi}\left(cos^{-1}(\nu/\nu_c) - (\nu/\nu_c)\sqrt{1 - (\nu/\nu_c)^2}\right)$ (Rowlands, 2017)."

Added a reference to equations 1 and 2 in the caption for Fig. 3.

row 148 "the diffraction limit decreases with smaller apertures"
--> "the diffraction limit decreases with smaller apertures (f-numbers)"
ln 150. Added "(large f-numbers)."

row 149 "1/lambad N" should be "1/(lambad*N)".
ln 150. Corrected to "$1/(\lambda N)$" here, and in the caption for Fig. 2.

row 175-180 "more visible as the size of the Airy pattern"
"the distance between the centers of two disks is equal to their radius"
"when the circle of confusion reaches a size of twice the pixel pitch"
--> I would welcome that every time it is clearly specified if you mean diameter or radius (like in the 2nd quote).
Agreed. We replaced "size" with "diameter" throughout the text, and simplified Eq. 2 by:
(1) directly using the Airy diameter (2.44) instead of the radius (1.22 × 2) and,
(2) replacing the term for aperture diameter ($D$) by the equivalent expression ($f/N$).

Equation 2
"GSD", which is not the resolution of the image, is a well defined term (your equation 1).
Thus I object against introducing the term "diffraction limited GSD" in equation 2, because any later mentioning of GSD creates confusion whether you mean really "GSD" or use it as a short for "diffraction limited GSD". And, indeed, in the caption of Fig. 3 you write "Diffraction limited ground sampling distance (GSD)" which is ambiguous in the way that there GSD might refer only to "ground sampling distance" or the whole term "diffraction limited ground sampling distance". In the end of that caption you also mention another term: "theoretical GSD". Furthermore, later, you refer to equ.2 as circle of confusion (which I think is better). Any term that really refers to the resolution (instead of the sampling = GSD) should by clearly discernable. Some authors use the term "GRD" (ground resolved resolution). Although, that term itself leaves open what effects are considered (just diffraction, or even the full point spread function), it still means obviously something else than GSD.
This is a good point. To make clear the distinction between sampling and resolution, we replaced "diffraction limited GSD" with "GRD" throughout the text, and added the following sentence (lns 177–8):
"In aerial (and satellite) photography, ground resolved distance (GRD) refers to the smallest resolvable detail on an image, given the limitations of the imaging system, including diffraction effects."

row 202 "Wide angle lenses exhibit negative (barrel) distortions which present as decreasing image magnification from the center of the frame towards the edges, while positive (pincushion) distortions are characteristic of telephoto lenses (70 mm or above)."
--> This is opposite to my experience which is:
1. That the sign of the radial distortion is not predictable from the angle of the camera.
2. The longer the focal length the more the lens shows no distortion at all.
Admitted, this is my personal experience, but currently, the quote is without reference. So, please, either provide a reference, or rephrase.
(On row 197 you write "the downside being that short focal lengths are more prone to distortions", if this is inverted to "longer focal lengths are likely to have less distortion", it fits to my experience.)
In general, distortions vary from barrel to pincushion as focal length increases from one extreme to the other, going from wide-angle to telephoto. This is often more evident for zoom lenses which show both types of distortion (e.g., Ray, 2002). Overall, however, the amount of distortion is more pronounced in lenses with short focal lengths and tends to be less visible with long focus lenses. In that sense, we agree with your comment. To clarify this point, we updated the text and added a reference to Ray (2002) (ln 215–8):

"Zoom lenses tend to display more complex distortions and a combination of both types, transitioning from positive to negative with decreasing focal length (Ray, 2002). The amount of distortion corresponds to the difference between the real image and the theoretical (undistorted) one, often reported as a percentage of image height, and is generally less pronounced in long focuses lenses but intensifies with increasingly short focal lengths (Ray, 2002)."

 Fig. 4

I am not sure, if (a) and (b) are derived from the very same RAW image, or only (a); and (b) is a direct JPG from the camera (and thus in principle a different photograph than the RAW for (a))?

They are versions of the same image, saved in-camera as both RAW and JPEG format. We charged the labels on the figure from "TIFF" to "RAW" to avoid confusion and rephrased the caption:

"Figure 4. Exposure adjustments performed on two versions of a single underexposed image captured in (a) 14-bit RAW, and (b) 8-bit JPEG formats. Both files were saved in-camera and imported into Lightroom for editing. Stronger adjustments are required for the JPEG (b1) to reach a comparable overall level of exposure and retrieve an equivalent amount of information to the RAW image (a1). With more extreme adjustments, the RAW image (a2) remains useable, while on the JPEG (b2) some information is lost in the darker shadows (bright blue pixels) and compression artefacts and false colour (purple patches) combine to degrade image quality. The RAW images were subsequently exported as 16-bit TIFFs for further processing."

row 250 "Including the affinity and non-orthogonality coefficients in the camera calibration matrix at the image alignment stage should partially compensate for this effect."

--> This works only (or the more) the flight speed is constant and the terrain is flat. You may wish to add/clarify that.

Added the following text to clarify (lns 262–3):

"... should partially compensate for this effect, although it is less likely to be effective with large and rapid changes in flight speed, direction, and height above ground."

Remark 1: (Again from my experience) affine parameters need to be introduced per image (and not per camera).

In theory, this would be our assumption as well, especially with large or rapid variations in camera motion. However, when dealing with our two datasets we found that letting these two coefficients vary led to over-fitting, giving unrealistic values for the focal length and principal point coefficients. Ultimately, using the 8-parameter camera model (omitting b1-b2) improved our results.

Remark 2: The developers of Pix4D have a paper about their method on rolling shutter compensation, which is better than the affinity in image space as it directly works on the change of the exterior orientation parameters per image:

https://s3.amazonaws.com/mics.pix4d.com/KB/documents/isprs_rolling_shutter_paper_final_2016.pdf

Thank you for bringing this useful paper to our attention. We have now added more information on the rolling shutter compensation to the text, along with a reference to the Vautherin et al., (2016) paper (lns 263–9):

"Various software, including Pix4D and Agisoft Metashape, have also implemented camera models to compensate for rolling shutter effects, estimating camera motion (translation and rotation) during exposure and modelling external orientation parameters per row of pixels on the sensor (instead of per image) (Vautherin et al., 2016). However, the performance of the correction is also sensitive to survey configuration, showing better results with more regular gridded flight patterns at relatively constant speed, and especially when combining nadir and oblique images. When correcting for rolling shutter, simultaneously solving for the affine distortion parameters has been shown to degrade accuracy due to an overparameterisation of the model (Zhou et al., 2020)."

row 254 "including" --> "included" (?) OK
row 266 "0.43" --> "4.3" OK

row 280 "The direct georeferencing method ... similar precision to the ground-based approach where camera position information is acquired with multi-frequency survey-grade GNSS equipment"

--> Does this last part ("where camera position …") refer to the ground-based approach?

No, it refers to the direct georeferencing approach. We have moved the last part of the sentence higher up to avoid confusion (lns 297–300):

> "The direct georeferencing method, using airborne control measurements, represents a major logistical advantage for aerial surveys in remote locations as it eliminates the need for a network of GCPs and, where camera position information is acquired with multi-frequency survey-grade GNSS equipment, it has been shown to produce results of similar precision to the ground-based approach."

Remark: A photogrammetric survey that fully relies on direct georeferencing using GNSS, and thus without a single GCP, is prone to deliver results with a large height bias. Because if the camera calibration is considered unknown and thus is estimated during the bundle block adjustment, then any small bias in the estimated focal length causes a large height offset. This is especially true for vertical images, and may be mitigated using oblique images. As you mention, this is predominantly an issue for nadir datasets, and especially for surveys over relatively flat terrain. Combining oblique images with nadir datasets has been shown to reduce GCP requirements (e.g., Nesbit and Hugenholtz 2019). Fortunately, our datasets are composed of oblique images captured in a convergent geometry, which minimises this issue: in our results, using no GCPs, RMS errors at checkpoints were roughly twice as high in the vertical than horizontal, but all were <0.7 m.

General comment to the theory (section 2): Interestingly, the problem of depth of field is completely neglected. Although, its importance increases with smaller viewing (focus) distance, it belongs into this theory part. Actually, you refer to defocus in row 591. Especially, the hyperfocal distance would be an interesting feature maybe not known to everybody of the target audience.
Thanks for pointing this out. We have added the following text describing the problem of depth of field and hyperfocal distance in section 2.2.2. (lns 203–10):

> "Focal length and aperture also define the hyperfocal distance, corresponding to the focus distance giving the maximum the depth of field (DOF), defined as the zone of acceptable focus. The hyperfocal distance decreases with focal length and aperture, with wide-angle lenses and large f-numbers maximising the DOF. Focusing a lens at infinity places the near edge of the DOF at the hyperfocal distance (Ray, 2002) which, for an effective focal length of 24 mm at f/5.6 is 3.4 m, meaning everything falling any further will be acceptably sharp. Further closing the aperture to f/11 reduces the hyperfocal distance by about half (to 1.7 m) but also impacts system resolving power by increasing diffraction softening. In aerial photography, where the height above ground exceeds the hyperfocal distance and DOF is not a concern, selecting an aperture minimising diffraction and motion blur is preferable."

page 14/15: I am a bit confused regarding your "off-nadir" images. On row 365 you say ""$>5°$" and on row 372 "30-50°". Why use these rather different definition thresholds, and not just provide some information on the off-nadir angles your two sites used; e.g. 5th and 95th percentile, and add that info to table 2. Additionally, it is not clear in which direction the off-nadir angle is applied; as pitch or as roll, or something between?
Yes, this can be confusing. The definition varies between studies, but 5° seems to be a minimum to qualify an image as off-nadir, while 30–50° are more typically used to describe oblique aerial photography (e.g., Nesbit and Hugenholtz 2019), and closer to the angles in our two studies here. With the variable topography, the camera handheld, and without additional attitude information, it is difficult to give more precise estimates of the actual angles.

Following your suggestion, we replaced the general definition ">5° off-nadir" with the approximate angles used in the two surveys, and added a sentence specifying that the camera was pointed roughly 30–50° off-nadir, predominantly in the roll direction (lns 397–9):

> "In both surveys, the camera sensor was oriented with the short edge (vertical) parallel to the direction of aircraft travel (yaw 0°), and the viewing direction roughly orthogonal to the flightpath (pitch 0°), between 30–50° off-nadir to the right at TF (roll >0°) and to the left at ED (roll <0°)."

We also clarified the distinction between simply "oblique" and "convergent" imagery (lns 389–90):

> "... with a convergent image geometry with varying angles oriented around a central area of interest showing the biggest improvement (Sanz-Ablanedo et al., 2020)."

Here it would be super helpful to include the viewing direction in Fig. 5 for every 50th image or so.

In table 2 further some info on the GSD would be interesting to get some idea. I know with oblique angles and variable terrain there is no straight forward way to come up with a representative value, but currently there is no mentioning at all.

To help clarify this, we added a line in Table 2 with a range for the GSD based on the altitude above ground level information in the same table, also specifying the following in the caption:

"The ground sampling distance (GSD), based on the indicated aircraft altitude a.g.l., represents an upper-bound estimate assuming nadir imagery."

Adding the viewing direction to Fig. 5 would make it very busy (especially the top panel) and harder to read, but we added information on the viewing direction in the caption to help with this:

"The camera viewing direction was roughly orthogonal to the direction of travel (indicated by an arrow), looking left out of the aircraft in (b), and right in (c)."

row 392 You took pictures through the front passenger window. It would be interesting to know how the camera calibration was effected by that (in comparison to the other data set).

ln 414. Clarified that pictures were taken "out of an open window on the front passenger side." No extra glass was placed in front of the lens that could have affected camera calibration.

row 413 "PPP" is mentioned here the first time, please, add a reference.

ln 437. Added a reference to Kouba and Héroux (2001) and Kouba et al., (2017).

row 443 Here and elsewhere you mention an aperture of "f/5". Although, such value is possible in principle, it would be very unusual, because the usual f-stop numbers are the powers of sqrt(2), the closest being thus 4 and 5.6, which you also have in Fig. 2+3.

That is correct, f/5 (used in the EF survey) is one-third of a stop above the full stop of f/5.6 (used in the TF survey). While f/5 might not be particularly common, we are not aware of a reason not to use one half- or one third-stop increments in photogrammetry. However, this is still a good point, and so we updated ln 676 in the final recommendations from "aperture of f/5.0–5.6 ...", to:

"Use an aperture two to three stops up from the maximum, ideally around f/4.0–5.6 ..."

row 456 "at or below the size of the circle resulting from diffraction"

--> you mean "diameter"? Also add a reference to the equation in the theoretical part.

lns 480–1. Replaced "size" with "diameter" and added a reference to Eq. 2 (GRD).

row 478 "Lastly, a variable exposure gain was automatically applied to all images to brighten underexposed areas and match total exposure of successive images."

--> Can your provide a bit more information on how this exposure gain is controlled?

To provide more information on the variable exposure gain, we replaced the above sentence and the next one with the following text (lns 502–7):

"Lastly, a variable exposure gain was applied to all images to brighten underexposed areas and increase the level of detail and available information for feature extraction. Here, Lightroom automatically adjusts the total exposure (EV) of successive images captured with different in-camera exposure settings (i.e., shutter speed and ISO), to match a selected reference image. This was performed in batches, selecting overlapping images with similar content to that of the reference image, to even out differences in illumination between images and enable a more uniform orthophoto reconstruction."

row 486 "the most time-intensive task in post-production is masking extensive swaths of sky and any terrain beyond the area of interest."

--> What would have happened if you would not have done this masking? Would the bundle block adjustment have completely failed for all images, or just for the affected images? Or would the adjustment have worked, but the

subsequent dense point cloud extraction would have failed (if so, why not simply define a bounding box prior to deriving the dense point cloud)?

Without masking the sky, the bundle adjustment still works, and most of the bad tiepoints can be filtered using gradual selection. The main issue then is the large amount of noise in the dense point cloud. Because on some photos the glacier surface meets the sky (or some distant mountain), defining a bounding box to exclude the background is virtually impossible. The masks are then useful to exclude part of images from the depth maps, and so from the dense cloud.

General comment on the bundle block adjustment in Metashape:
- What accuracies were assumed for the GNSS image positions?
- What GNSS residuals were obtained after the adjustment?
- What reprojection error was obtained?

Since the focus here is on the acquisition and optimisation of source data, not on the photogrammetry processing itself, this detailed information is included in the PhD thesis that this paper is based on (Medrzycka, 2022).

row 575 "Ideally, horizontal accuracy should be higher or equivalent to the spatial resolution of the final gridded products. Here, both DEMs and orthomosaics were gridded at 0.5 m resolution and horizontal checkpoint misalignment errors remain below that level for both reconstructions."
--> I never heard of this rule and, actually, do not subscribe to it. The gridding of the results (DEM from dense image matching and orthomosaic) should fit to the image resolution, i.e. the finest details in the image should also be included in these results, independently of the spatial accuracy of the georeferencing. Even if accuracy and grid width do not match, as in your case, the results provide valuable information about these fine details, however, you only are able to derive the location with a certain limited accuracy. For that reason, the accuracy always should be communicated together with the results.

The gridding was done independently of the accuracy of the results. What we mean here is that ideally the horizontal accuracy would be better than the resolution so as not to degrade the quality of the final product. In this case, final resolution was 0.5 m and RMS errors for the checkpoints are given in the previous sentence (lns 600–1).

In more general terms, when GSD is better constrained, we would aim to minimise the ratio of RMSE/GSD, which is the same as minimising RMSE/DEM and orthophoto resolution since final pixel size should be roughly equivalent to the GSD.

Furthermore, the error in the georeferencing is usually a global one, meaning that your result could be improved simply by shifting in 3D to obtain a much better georeferencing. The latter can be done even on the results themselves; e.g. in case of time series where one epoch (with the best georeferencing quality) serves as reference (and enough corresponding stable areas are present, of course). Thus it would be a waste of potential to set the gridding of the result to the obtained georeferencing accuracy.

Assuming survey-wide systematic error only, simply shifting the reconstruction could indeed improve uniform registration accuracy, but this is not the case with spatially-uncorrelated random error as we had in our study.

Last not least, photogrammetry works even without GCPs and GNSS (one only needs a known distance for scaling). In this case no meaningful (absolute) georeferencing accuracy can be obtained, but the resolution of the images still serves as a guide. As you outlined in the theory the actual resolution of the images is not easily determinable, because it depends on so many factors. However, the GSD is well defined and easily obtainable (at least in case of nadir images over "flat" terrain). So the usual way in photogrammetry is to adopt the GSD as gridding value for the DEM and the orthophoto.

In your case I am not sure what the GSD is, but from the given point density values of around 15 pts/m$^2$, we see that the average distance between these points is 1/sqrt(15) = 26 cm. Thus you could at least create your DEM and the orthomosaic with 25 cm grid width and should thus be able to get a bit more out of your results than compared to the chosen 50 cm (provided the images were not dramatically effected by blur). I could imagine, that the orthomosaic could be created with an even smaller pixel size, because with dense image matching only in optimal cases one really gets a 3D point per image pixel. Furthermore, in your case, you will have a high variability of the

GSD, and in order to get the details even in the images with the smallest GSD, one thus could base the orthomosaic pixel size not on the average GSD but some smaller statistical value, like the 10th percentile.
We agree with all of the above and we initially did build the DEMs with ~0.25 m grid spacing (based on the average point density relationship you mentioned), but any smaller cell size resulted in patchier reconstructions. Ultimately, 0.5 m was chosen as a compromise between processing time and level of detail required by the specific project. In this case, the final products were compared with much lower resolution reconstructions from historical aerial photographs. Working with the DEM and orthophotos at higher resolution was computationally heavy and provided very little improvement (if any) to the final results.

We updated the text and added more information to clarify this point (lns 601–8):
   "Ideally, horizontal accuracy would be higher or equal to the spatial resolution of the final gridded products which, for surveys with more regular geometry and constant height above ground, should be roughly equivalent to the GSD. In. In this case, where GSD is not easily constrained, point density is useful to define an appropriate pixel size for the gridded products. Here, DEMs and orthomosaics were gridded at 0.5 m, or roughly half the achievable resolution based on the average point spacing of 0.27 m at TF, and 0.22 m at EF. The 0.5 m represents a compromise between processing time and resolution and, in this case, is sufficiently detailed to answer the requirements of the specific project. Horizontal checkpoint misalignment errors remain below the 0.5 m cell size for both reconstructions."

row 665 "Due to data gaps, 28 cameras from the 10 Hz EF survey were disabled (~5 %), compared to 129 cameras (or 13 %) from the 15 s TF survey."
--> Here you use the wrong terminology from Metashape. You mean 28 and 129 "images" not "cameras". Actually, here do you mean that the entire images were disabled, or that the GNSS locations were disabled (due to big interpolation error)?
(Actually, Metashape should be able to link images without GNSS location information to their neighboring images (with GNSS location), provided the image content allows for enough feature points.)
That's correct, Metashape can align images with no valid coordinates. The 28 and 129 images were still used, but the associated camera position estimates were ignored in the bundle adjustment (i.e., disabled). In the original sentence, the term "cameras" comes from the "Reference" tab in Metashape which lists the imported photos in the "Cameras" column, along with the camera position coordinates. In this context, the terminology seems to be correct, although we agree that the wording is confusing, so we changed it to (lns 723–5):
   "Due to data gaps in the GNSS observations, camera position estimates for 28 images (~5%) from the 10 Hz EF survey, and 129 images (or 13%) from the 15 s TF survey, were marked as invalid and were omitted from the SfM workflow."

It would be interesting to list the numbers of images regarding: originally taken vs. disabled images (classified for whatever reasons (e.g. blur)).
We have added more information to Table 2 with the number of used images and those with valid coordinates (also corrected a mistake with the total number of images from the TF survey on lns 405 and 508).

Finally, a general comment: If you cite a book of several hundreds of pages, then please include the page number in the quotes; e.g. Rowlands, 2017.
The norm in the geosciences, and associated style guides, is to only provide a page number when text is directly quoted from a book or paper (e.g., https://blog.apastyle.org/apastyle/2015/03/when-and-how-to-include-page-numbers-in-apa-style-citations.html). We have therefore not included page numbers with our references, but would be happy to do so if the editor agrees with the reviewer's suggestion.

---

## Author Comment (AC2)

**Reply to RC2**

In this paper, the authors have provided a thorough overview of the technological and survey design considerations that impact the usability and accuracy of photogrammetric surveys and derived products. The authors then compare these considerations to the results of two prior surveys made on remote glaciers in Arctic Canada. I found the first "half" of the paper to be very well put-together and easy to read, which is no small feat for such a technical subject!

Because the surveys were (presumably) made before the start of the PhD that this work comes out of, it feels as though there is a slight disconnect between the review of the first half and the discussion in the second half - there isn't much indication that the lessons learned from the review informed the design of the field surveys, or that these lessons were "tested" by comparing the results of different approaches. This is entirely understandable, but I think that some additional critical reflection about how the surveys might have been improved would help to bridge this gap.

Thank you for the useful feedback. It's correct that the two parts of the study were undertaken at somewhat different times during the PhD, but we hope that the combination of a theoretical review combined with measurements under real-world conditions help to provide insights for how to optimize air photo surveys with field conditions that are often not as planned. To do this, we outline the weaknesses and discuss aspects that were improved or adjusted between the first and second survey, and those that should be improved in the future, but testing every factor, its impact on image quality and on the results of the bundle adjustment, is beyond the scope of this study. With multiple interacting factors contributing to the accuracy of topographic data, separating the impact of individual aspects is not straightforward, but we agree that it would be interesting to investigate in future work.

In response to the reviewer's comments, we have added a paragraph to the conclusion stating the main way the surveys could have been improved (lns 757–67):

> "Beyond any challenges inherent to the survey areas, logistical time constrains, and weather delays, the outcomes of the two surveys presented in this study were adversely impacted by unforeseen complications from system malfunctions (GNSS receiver defaulting to low logging rate) and human error (forgetting the intervalometer, accidental loss of focus). However, the main weakness was in the positioning performance being limited by the GNSS system which, despite consisting of dual-frequency, survey-grade components, was only able to record GPS observations (i.e., from a single satellite constellation). With continuous developments in GNSS technology, and multi-constellation/frequency equipment becoming increasingly accessible, upgrading the positioning system used here will likely result in fewer data gaps and significantly improve direct georeferencing accuracy in future surveys. Additional INS data could further help derive more precise control measurements taking into account variations in aircraft attitude. When coupled with precise synchronisation with the camera sensor, INS-aided GNSS navigation could help reduce uncertainties in camera position estimates and ensure more accurate topographic reconstructions."

Obviously, additional surveys are not feasible, but "simulation" of different settings after the fact (using ETTR images by shifting the histogram, using JPG instead of RAW images, not correcting chromatic aberrations, etc.) might be possible and give some indication of what impact these different choices have on the final product. This would help to solidify the connection between some of your proposed recommendations and the final results, though I only suggest doing this if it is easy to do - if not, some additional comparison to literature that has reviewed the impact of these different design choices is more than enough.

Unfortunately it's not possible to simulate the different settings as suggested (and access to our field sites is very difficult to repeat any surveys), so instead we have followed the reviewer's suggestions and included additional references to studies demonstrating the impact extreme lens distortions (Thoeni et al., 2014; Girod et al., 2017), rolling shutter effects (Vautherin et al., 2016; Zhou et al. 2020, Stark 2021), survey configuration and image capture geometry (Sanz-Ablanedo et al., 2020).

We also include an example from out dataset to illustrate the impact of image resolution on feature detection and tiepoint density which studies have shown to be directly related to result accuracy (lns 630–40):

"Maximising image information content enhances feature detection and matching performance, resulting in higher point density, and significantly improving both reconstruction accuracy and precision (Gienko and Terry, 2014; Mosbrucker et al., 2017). In this study, feature detection was performed after upsampling all images by a factor of four. Increasing image resolution involves interpolating pixel values which can introduce artefacts (Rowlands, 2017), and is therefore only advantageous for high resolution and high quality images where it can be 635 helpful with identifying more features and matching them with sub-pixel precision. Tested on a subset of 20 images from the EF dataset, the average number of features detected on upsampled images was 7–8 times greater than with the original resolution data (1.6 vs 0.2 million points per image), and the average number of valid matches increased by a factor of four (310,000 vs 75,000 points per image). The total point count in the corresponding clouds was five times greater based on the upsampled data (2.3 vs 0.46 million points each). This underlines the importance of selecting high performance imaging hardware maximising 640 resolving power and image quality to ensure high point density, and in turn more accurate topographic reconstructions."

Another consideration that I think could be better explored is the minimum feasible design. Obviously, a number of studies obtain usable results, even with less ideal camera setups (e.g., Welty et al., 2013; Girod et al., 2017), so at least some part of survey design depends on the needs or application. Given how rare the "ideal" situation is, some additional thought/reflection on what the "minimum" settings (or the non-negotiable choices) should be in order to achieve a useful output would be useful here.

This is a good point, and in response, we have added a paragraph in the discussion referencing other studies that have demonstrated adequate results using what could be considered the minimum non-negotiable camera setup, as well their key considerations for improving the system beyond the bare-minimum (lns 618–40):

"The camera (Nikon D850) and lens (NIKKOR AF-S 24mm f/1.8G ED) used in this study were selected based on the key aspects discussed in Sect. 2.2, prioritising high resolving power, high SNR, and low geometric distortions. Low-cost compact cameras, including smartphones, action cameras, and cameras commonly mounted on consumer-grade RPAS, have been shown to provide adequate results for some applications, but comparisons with higher performance imaging systems show significant improvements in final outputs (e.g., Thoeni et al., 2014; Eltner and Schneider, 2015; Micheletti et al., 2015; Girod et al., 2017; Stark et al., 2021). With image resolution (GSD) being the main contributing factor to result accuracy, studies have also demonstrated the importance of maximising the level of detail captured (Mosbrucker et al., 2017), avoiding extreme distortions from very wide-angle (fisheye) lenses (Thoeni et al., 2014; Girod et al., 2017), minimising rolling-shutter effects with fast sensor readout speeds or global shutters (Vautherin et al., 2016; Zhou et al., 2020; Stark et al., 2021), and ensuring precise synchronisation between the camera and positioning system (Welty et al., 2013)."

We have also added a paragraph summarising the main priorities for survey planning and image capture settings (lns 663–72):

"Image capture and survey planning are tightly related to the size, topography, and surface characteristics of the study area. For a given camera and lens combination, flying height will determine the GSD (Eq. 1), which should be selected based on the project and required level of detail, while keeping in mind the extent of the survey area and the flight time and number of images needed to provide full coverage with sufficient overlap. Other aspects should be configured to maximise the GRD while ensuring adequate exposure, keeping diffraction effects (Eq. 2) and motion blur (Eq. 4) to a minimum, ideally at or below 1.5 pixels. Maintaining image sharpness is crucial and, in low light conditions, decreasing flight velocity will allow faster shutter speeds while also reducing distortions from rolling shutter effects (Eq. 3). With large survey areas or limited flight time, where decreasing flight velocity is not possible, or with variable wind conditions causing sudden jumps in aircraft motion, it is preferable to increase ISO at the cost of introducing some noise, rather than risking blurry images."

Specific comments
* * *
line 16: add "e.g." to Mölg and Bolch reference OK

line 67: why are helicopters preferred here? Can you explain the motivation/reasoning why this would be so? I would think that this will depend on the particular study area (size, remoteness), the available camera setup, and perhaps the prevailing weather conditions, but it's not clear to me why helicopters would be preferred in all cases. The main point is that remotely piloted aircraft have several limitations which make them unsuitable to survey entire glaciers in certain conditions. At the latitudes of our study sites, and given the size of the glaciers that we want to measure, those unfavourable conditions are the norm, making manned aircraft the only option in almost all cases. Examples of issues that we have encountered in the field when trying (and failing) to use UAVs are frequent crashes due to proximity to the magnetic north pole, cold temperatures that severely limit range, and glaciers that are orders of magnitude larger than be covered by a drone.

As to why we use helicopters over fixed-wing aircraft, it mainly has to do with the logistics of chartering a fixed wing aircraft specifically for a scheduled survey 100s of km away from their home base, versus the flexibility of fitting in a survey flight in an existing field schedule with the helicopter stationed in camp. Fixed wing aircraft in the Canadian High Arctic also typically don't have a camera hatch or any way of taking photos out of an open window while airborne, and they can't perform the sharp turns at low elevations that we undertook in our surveys.

Table 1: can you provide, in either the caption or the text, an explanation of "high" and "low"?
Since the distinction is in the pixel count, and unrelated to sensor format, we removed "high" and "low" from the table column and added the following text in the caption:
"The Nikon D850 fits in the FF category of newer sensors with relatively higher pixel count than the more standard 20–24 Mpx."

line 153: do you mean "stepping" instead of "stopping" here?
Here "stopping down" refers to decreasing the aperture by two stops, that is, increasing the f-number from f/4 to f/8. "stepping" up/down has another meaning and refers to fitting an adapter ring in front of the lens to use filters with a different thread size, but that isn't what we're referring to here.

line 165: perhaps "a combination" is a better word choice than "the product" to avoid confusion with the mathematical meaning of "product" - unless this is the meaning that is intended here.
"Product" is the correct word here, as multiplying the MTF of all separate components gives the system MTF.

line 206-7: I think this again depends on the needs of the study and equipment availability; it is possible to get usable results from "unsuitable" cameras.
Agreed. We added a paragraph in the discussion (lns 619–27) to underline this point (see the response to the last general comment above).

Figure 4: it might be useful to indicate the outline of (a2, b2) in (a1, b1). Added the outlines of a2/b2 in a1/b1.

line 359: insert space between "a.s.l." and "The" OK
line 530: remove stray "s" OK

References
* * *
Girod, L., Nuth, C., Kääb, A.M., Etzelmüller, B., Kohler, J., 2017. Terrain changes from images acquired on opportunistic flights by SfM photogrammetry. The Cryosphere 11, 827–840. https://doi.org/10.5194/tc-11-827-2017

Welty, E.Z., Bartholomaus, T.C., O'Neel, S., Pfeffer, W.T., 2013. Cameras as clocks. Journal of Glaciology 59, 275–286. https://doi.org/10.3189/2013JoG12J126

---

## Author Response (AR2)

We thank the editor and first reviewer for the follow-up on the revised manuscript. In this updated version, we have made the following corrections:

1. Added page numbers to references from books, namely for Ray (2002) and Rowlands (2017). Please note that previous references to the first edition of Rowlands (2017) were now updated to the second edition of Rowlands (2020).

2. Updated the caption for Fig. 4 and specified the source for the base-image as being 'from the Tanquary Fiord (TF) survey (3 August 2018)'.

3. Removed the term '(vertical)' (line 399 in the revised document).